# Submarine melt as a potential trigger of the North East Greenland Ice Stream margin retreat during Marine Isotope Stage 3.

Ilaria Tabone[1,2], Alexander Robinson[1,2], Jorge Alvarez-Solas[1,2], and Marisa Montoya[1,2]

[1]Universidad Complutense de Madrid, 28040 Madrid, Spain
[2]Instituto de Geociencias, Consejo Superior de Investigaciones Cientificas-Universidad Complutense de Madrid, 28040 Madrid, Spain

*Correspondence to:* Ilaria Tabone (itabone@ucm.es)

**Abstract.** The Northeast Greenland Ice Stream (NEGIS) has been suffering a significant ice mass loss during the last decades. This is partly due to increasing oceanic temperatures in the subpolar North Atlantic, which enhance submarine basal melting and mass discharge. This demonstrates the high sensitivity of this region to oceanic changes. Alongside, a recent study suggests that the NEGIS grounding line was 20-40 km behind its present-day location for 15 ka during Marine Isotope Stage (MIS) 3. This is in contrast with Greenland temperature records indicating cold atmospheric conditions at that time, expected to favor ice-sheet expansion. To explain this anomalous retreat a combination of atmospheric and external forcings has been invoked. Yet, as the ocean is found to be a primary driver of the ongoing NEGIS glaciers retreat, the effect of past oceanic changes in their paleo evolution cannot be ruled out and should be explored in detail. Here we investigate the sensitivity of the NEGIS to the oceanic forcing during the last glacial period using a three-dimensional hybrid ice-sheet-shelf model. We find that a sufficiently high oceanic forcing could account for a NEGIS ice-margin retreat of several tens of km, potentially explaining the recently proposed NEGIS grounding-line retreat during Marine Isotope Stage 3.

## 1 Introduction

The Northeast Greenland Ice Stream (NEGIS) is the largest ice stream in the Greenland Ice Sheet (GrIS), extending more than 600 km inland (Joughin et al., 2001) and discharging 12% of the whole ice sheet through three outlet glaciers (Rignot and Mouginot, 2012): Nioghalvfjerdsfjord Gletscher (79N), Zachariae Isstrøm (ZI), and Storstrømmen Gletscher (SG), which is today a surging glacier. These marine-terminating glaciers have suffered huge changes in the last decades. In less than 15 years the ZI floating tongue has lost 95% of its size as a result of an enhanced mass loss (Mouginot et al., 2015). Concurrently, since 1999 the 79N ice shelf has lost 30% of its thickness at the grounding line (Mouginot et al., 2015), contributing to its inland retreat by 2 km (Mayer et al., 2018). However, since 79N is retreating over an upward-sloping bed (Mouginot et al., 2015), it may be less prone than ZI to an unstable retreat. This has been recently examined through an ice-flow model pointing out that its floating tongue has to lose several tens of km of ice before the glacier becomes unstable (Rathmann et al., 2017).

Enhanced stability of 79N has been recently tested under various future warming scenarios by another modelling study (Choi et al., 2017), suggesting that it may be related to the presence of pinning points (such as ice rises) near the calving front. Ice loss from these two marine-terminating glaciers is thought to be partly related to the increasing temperature of North Atlantic waters (Khan et al., 2014; Mouginot et al., 2015), which increases the oceanic heat flux and accelerates the submarine melting (Mayer et al., 2018). This hypothesis is supported by the three-decade-long observed warming in the subpolar North Atlantic (Straneo and Heimbach (2013) and references therein). Moreover, warmer oceanic waters in Fram Strait could directly reach the 79N, further increasing its basal melting and potentially causing the loss of its floating ice tongue (Schaffer et al., 2017). Although 79N has been suggested to be more resistant to increasing basal and frontal melt than ZI (Choi et al., 2017), new evidence revealing that both glaciers retreated beyond their PD margins during the Holocene indicates that this conclusion may be too conservative (Larsen et al., 2018).

Reconstructions suggest that during the Last Glacial Maximum (LGM), ca. 21 ka Before Present (BP), the northeastern region of the GrIS considerably advanced at 250-300 km from the present-day coastline, likely reaching the continental shelf break (Arndt et al., 2015, 2017; Evans et al., 2009; Winkelmann et al., 2010). Although the age of these LGM reconstructions is still poorly constrained, the combination of cosmogenic exposure and radiocarbon dating has recently facilitated the reconstruction of the position of the NEGIS over the last 45 ka (Larsen et al., 2018). The paleo records emerging from this study, combined with a collection of geological data assembled in the last 20 years (Arndt et al., 2015, 2017; Bennike and Weidick, 2001; Evans et al., 2009; Weidick and Reeh, 1996; Winkelmann et al., 2010), suggest that its ice margin considerably fluctuated in magnitude throughout this period. Around 41-26 ka BP during Marine Isotope Stage 3 (MIS-3, ca. 60-25 ka) the NEGIS front was ca. 20-40 km farther inland than today, then advanced by more than 250 km toward the shelf break at the LGM and retreated again during the last deglaciation, at ca. 70 km behind its present-day position, where it stopped most of the mid- and late Holocene (7.8-1.2 ka BP). The Holocene retreat was likely due to an increase in both atmospheric and oceanic temperatures, whilst the retreat during MIS-3 was attributed by Larsen et al. (2018) to a combination of atmospheric and external forcings. However, the potential role of oceanic forcing in this retreat has not been explicitly investigated from a modelling perspective. In the light of the ongoing changes in the GrIS attributed to ice-ocean interactions, this appears as a plausible mechanism that needs to be investigated. Moreover, since it is expected that warmer Atlantic waters entering the fjords will strongly affect the NEGIS margin in the future, assessing its response to similar past warm oceanic conditions will provide new insights into the future stability of its glaciers front.

Here we use an ice-sheet-shelf model to investigate the sensitivity of the NEGIS grounding line to changing oceanic conditions during the last glacial period. The submarine melting at the grounding line is parameterised in such a way that basal melt is allowed during relatively warm time periods such as the present, the last interglacial (LIG, ca. 128-116 ka BP) or MIS-3, whereas it reaches zero at the onset of the LGM. We study the NEGIS marine margin response to increasing basal melting rates during MIS-3 to show that a sufficiently high oceanic sensitivity could have driven a considerable NEGIS grounding-line retreat during MIS-3 from its former glacial position.

## 2 Methods

To simulate the NEGIS response to past oceanic forcing, we use the three-dimensional, hybrid ice-sheet-shelf model GRISLI-UCM (Alvarez-Solas et al., 2019; Tabone et al., 2018), adapted from the extensively used GRISLI model (Ritz et al., 2001). Grounded, slow-moving ice-sheet regions and floating shelves are treated through the shallow-ice approximation (SIA) and shallow-shelf approximation (SSA), respectively. In the transition between these two regimes (i.e., fast moving, grounded ice), the dynamics is solved by the simple addition of the SIA and SSA velocity solutions (Winkelmann et al., 2011). The SSA boundary condition is provided by basal sliding below the ice streams following a linear friction law, in which the basal shear stress $\tau_\mathrm{b}$ is proportional to the basal velocity $\boldsymbol{u}_\mathrm{b}$ and to a friction coefficient $\beta$ dependent on the effective pressure of the water at the base of the ice sheet $N_\mathrm{eff}$, as:

$$\boldsymbol{\tau}_\mathrm{b} = -\beta\,\boldsymbol{u}_\mathrm{b} \tag{1}$$

where

$$\beta = c_\mathrm{f}\,N_\mathrm{eff}. \tag{2}$$

The term $c_f$ depends on the characteristics of the bedrock topography (e.g. presence of sediments); $N_\mathrm{eff}$ is calculated as $N_\mathrm{eff} = \rho g H - p_\mathrm{w}$, where $\rho$ is the ice density, g the gravitational acceleration and $H$ the ice thickness. The sub-glacial water pressure $p_\mathrm{w}$ comes from a simple basal hydrological model based on a Darcy-type law, for which water flows at the base of temperate ice as driven by a gradient of hydraulic pressure. Despite the simplicity of this hydrology scheme, it provides a fair description of the outflow systems at the base of the ice sheet (Peyaud et al., 2007). Glacial isostatic adjustment of the bedrock due to variations in the ice load is reproduced through the Elastic Lithosphere Relaxing Asthenosphere model (Greve and Blatter, 2009). Unlike some recent hybrid models, the grounding-line position is defined through a pure flotation criterion involving ice thickness at the marine margin and prescribed sea level. Calving occurs whenever a two-constraint thickness rule is satisfied at the ice-ocean interface (Colleoni et al., 2014): first, the ice-front thickness must be lower than a fixed threshold (H=200 m here); second, the upstream ice advection does not succeed in preserving the ice-front thickness above that threshold.

The atmospheric temperature forcing applied to the model follows an anomaly method according to which the present-day climatological temperature $T_\mathrm{clim,atm}$ is perturbed by past anomalies obtained from a spatially-uniform proxy-derived index $\alpha(t)$:

$$T_\mathrm{atm}(t) = T_\mathrm{clim,atm} + (1 - \alpha(t))(T_\mathrm{LGM,atm} - T_\mathrm{PD,atm}) \tag{3}$$

The $\alpha(t)$ index is derived from the Greenland temperature reconstruction for the Holocene (Vinther et al., 2009), the North Greenland Ice Core Project (NGRIP) reconstruction for the last glacial period (Kindler et al., 2014) and the North Greenland Eemian Ice Drilling (NEEM) reconstruction for the LIG (NEEM, 2013), as in Tabone et al. (2018). The composed signal is then smoothed so that the spectral components below orbital frequencies are removed (i.e., periods below 16 ka). By construction,

$\alpha = 1$ at present day (PD) and $\alpha = 0$ at the LGM. $T_{\mathrm{clim,atm}}$ is taken from the regional climate model MAR forced by ERA-Interim (Fettweis et al., 2013), averaged over years 1981-2010. $T_{\mathrm{LGM,atm}} - T_{\mathrm{PD,atm}}$ is the glacial minus present-day (meaning preindustrial) atmospheric temperature anomaly simulated by the climate model of intermediate complexity CLIMBER-3$\alpha$ (Montoya and Levermann, 2008). The precipitation field is obtained following a similar approach based on the ratio of LGM and present-day precipitation, scaled by $\alpha$(t), as:

$$P_{\mathrm{ann}}(t) = P_{\mathrm{clim,ann}} \cdot \left( \alpha(t) + (1 - \alpha(t)) \cdot \frac{P_{\mathrm{LGM,ann}}}{P_{\mathrm{PD,ann}}} \right) \tag{4}$$

where $P_{\mathrm{LGM,ann}}$ and $P_{\mathrm{PD,ann}}$ are the LGM and PD annual precipitation provided by the same climate simulations as $T_{\mathrm{LGM,atm}}$ and $T_{\mathrm{PD,atm}}$. This approach has been adopted by many ice-sheet models to represent transient past precipitation when they are not coupled to a climate model (e.g. Banderas et al. (2018); Charbit et al. (2002, 2007); Colleoni et al. (2014); Marshall and Peltier (2000, 2002); Marshall and Koutnik (2006); Philippon et al. (2006); Zweck and Huybrechts (2005)). Surface ablation is calculated by the simple positive degree (PDD) scheme (Reeh, 1989). Although this method does not account for past insolation changes, since here we primarily investigate the sensitivity of the NEGIS to the oceanic forcing during glacial times, the choice of this melt scheme should bring only second-order effects to the overall results of this work.

The oceanic forcing is prescribed at the grounding line through a parameterisation of the submarine melt rate based on an anomaly method for which the present-day melt rate is perturbed by its past changes associated with variations in the oceanic temperature (Tabone et al., 2018):

$$B_{\mathrm{m}}(t) = B_{\mathrm{ref}} + \kappa \Delta T_{\mathrm{ocn}}(t) \tag{5}$$

where $B_{\mathrm{m}}(t)$ is the melt rate at the grounding line at a given time ($m\ a^{-1}$), $B_{\mathrm{ref}}$ is the present-day basal melting rate at the grounding line ($m\ a^{-1}$) and $\kappa$ is a coefficient representing the heat-flux exchanged between water and ice at the ice-ocean front ($m\ a^{-1}\ K^{-1}$). Past oceanic temperatures below the ice ($\Delta T_{\mathrm{ocn}}(t)$) evolve as:

$$\Delta T_{\mathrm{ocn}}(t) = (1 - \alpha(t))(T_{\mathrm{LGM,ocn}} - T_{\mathrm{PD,ocn}}) \tag{6}$$

where the $\alpha(t)$ index is that of Eq. 3 and $T_{\mathrm{LGM,ocn}} - T_{\mathrm{PD,ocn}}$ is the glacial minus interglacial oceanic temperature anomaly (K). The system of equations 5-6 can be solved assigning values to $B_{\mathrm{ref}}$, $\kappa$, $T_{\mathrm{LGM,ocn}}$ and $T_{\mathrm{PD,ocn}}$. However, some simplifications can be considered during the parameter assignment. Following Tabone et al. (2018), the reference melting rate $B_{\mathrm{ref}}$ is proportional to the oceanic sensitivity $\kappa$, as it is defined as

$$B_{\mathrm{ref}} = \kappa(T_{\mathrm{clim,ocn}} - T_f). \tag{7}$$

$T_{\mathrm{clim,ocn}}$ is the climatological mean of the oceanic temperature considered at the grounding-line depth (K) and $T_f$ is the freezing point temperature at the grounding line (K). The former is depth-dependent; the latter also depends on the distribution of salinity in the water column. Introducing $B_{\mathrm{ref}}$ in the equation is a simplification made to avoid the choice of values to be assigned to these two variables, that might be challenging and unconstrained (Beckmann and Goosse, 2003). For the sake of simplicity, $T_{\mathrm{clim,ocn}} - T_f$ can be considered as spatially (horizontally and vertically) constant, in the way that $B_{\mathrm{ref}}$ is defined

to scale directly with $\kappa$. Here, we prescribe $T_{\text{clim,ocn}} - T_f = 1$ K, thus $B_{\text{ref}} = \kappa \cdot 1$ K. Also, the glacial-interglacial temperature anomaly $T_{\text{LGM,ocn}} - T_{\text{PD,ocn}}$ is considered here to be spatially uniform and set to a value of -1 K (Annan and Hargreaves, 2013; MARGO, 2009). With these simplifications, the system of Eq. 5-6 is thus reduced to a problem of one degree of freedom ($\kappa$). Investigated values of $\kappa$ range from 0 to 10 $m\ a^{-1}\ K^{-1}$; thus $B_{\text{ref}}$ ranges from 0 to 10 $m\ a^{-1}$. These $\kappa$ values are consistent

with the inference from the Antarctic Ice Sheet that a change of 1 K in the oceanic temperature varies the melt rate by 10 $m\ a^{-1}$ (Rignot and Jacobs, 2002). Moreover, the resulting $B_{\text{ref}}$ values are in the range of the submarine melt observed at the grounding line of PD Greenland glaciers that have floating ice shelves (Wilson and Heimbach, 2017). Melting at the base of the ice shelves is defined to be the 10% of that calculated at the grounding line which reflects the decrease of melting rate observed towards the ice shelves (Anhaus et al., 2019; Münchow et al., 2014; Rignot and Steffen, 2008; Wilson and Heimbach, 2017).

However, this decrease is not parameterised here as a function of the distance from the grounding line. Instead, submarine melt is assumed to have a binary behavior: it is equal to $B_m$ at the grounding line and to the 10% of $B_m$ at all floating grid cells. Since the submarine melting rate at the grounding line is calculated to be spatially constant along the whole domain, the resulting value of the sub-shelf melt rate is also spatially uniform and is shared by all the ice-shelf grid cells of the domain. Note that refreezing below the grounding line is not allowed and it is cut off to zero, thus there is neither melting nor refreezing

during the LGM for the whole set of experiments, which is probably a simplification of reality. Melting and refreezing may vary strongly at local scales, as we know from present observations in Antarctica (e.g. Rignot et al. (2013)) and Greenland (e.g. Wilson and Heimbach (2017)). However due to the lack of data for basal melt along the NEGIS margins for the last glacial and the coarse resolution of our model (10 km), this assumption is considered as the most reasonable approach. The spectrum of resulting submarine melt rates leads to 11 different configurations, for which an increase in the oceanic sensitivity entails an

increase of the melting rate during MIS-3 (Fig. 1). These configurations allow investigating the role of the submarine melting rate on the NEGIS margin position during the last glacial period. Model simulations of the whole GrIS are initialized at 250 ka BP using the PD GrIS topography from Schaffer et al. (2016) and run under transient climatic conditions for two full glacial cycles. The first glacial cycle has been considered as a spin up. The analysis of the results focuses on the NEGIS sector.

## 3   Results

We calculate the grounding-line distance from the PD position on 48 transects intersecting the ZI and the continental shelf break (Fig. 2). Then we average the results to create one transient evolution for the grounding line for each oceanic forcing. The experiment with submarine melt prescribed to zero ($\kappa = 0$, $B_{\text{ref}} = 0$), which is hereafter referred to as the unperturbed experiment, shows the NEGIS margin rapidly advancing towards the continental shelf during glacial inception (Fig. 3). In less than 20 ka after the peak of the LIG, the grounding line advances through the inner sector of the continental shelf, extending

offshore to a distance of about 250 km from the PD NEGIS margin at around 65 ka BP. During MIS-3, the ice-margin position gradually advances towards its maximum glacial extent, which is reached at about 20 kyr BP (LGM), when the ice sheet becomes grounded at a mean distance of 40 km from the shelf break, reducing the area of the floating ice shelf in the region (Fig 4 a-e).

In all other simulations, the ocean forcing is switched on ($\kappa, B_{\mathrm{ref}} > 0$) and intensifies for increasing $\kappa$ (Fig. 1). The location of the grounding line at the LIG is the same in all simulations and thus insensitive to $\kappa$, and set mainly by the atmospheric forcing. Another common feature of these simulations is the response of the grounding-line position right after the peak of the LIG (Fig. 3): the inclusion of positive melt rates before 70 ka BP somewhat constrains the NEGIS margins 300 km upstream the grounding-line position obtained for the unperturbed experiment, remaining close to its LIG location. At about 70 ka BP, the ice margin starts to move towards the continental shelf break, stopping at ca. 170 km from its PD position after 10 ka of rapid advance.

The strongest reaction of the NEGIS grounding line to the applied submarine melting rate is found during MIS-3. By including a basal melt rate of $0-0.5\ m\ a^{-1}$ during MIS-3 ($\kappa = 1\ m\ a^{-1}\ K^{-1}$), the location of the NEGIS margin moves 100 km further inland with respect to the unperturbed experiment. Additionally, increasing the oceanic forcing not only helps to preclude the grounding-line advance (as compared to the unperturbed case with no oceanic forcing) but furthermore triggers its retreat. Submarine melt rates of $0-1.2\ m\ a^{-1}$ during MIS-3 ($\kappa = 4\ m\ a^{-1}\ K^{-1}$) constrain the NEGIS advance towards the continental shelf after glacial inception (ca. 116 ka BP) and trigger a slight inland grounding-line retreat by 80 km more which culminates at around 45 ka BP. A higher oceanic sensitivity ($\kappa = 5\ m\ a^{-1}\ K^{-1}$) leads to a further and earlier retreat during MIS-3. The minimum extent of grounded ice during MIS-3 is reached at around 50 ka BP, when the grounding line retreats by more than 100 km inland from its position simulated at 60 ka BP. The ice margin then remains steady until the end of MIS-3 (Fig. 3). This value of $\kappa$ and the resulting basal melt configuration (MIS-3 values above $1.6\ m\ a^{-1}$) act as a threshold above which the submarine melt rate forces the grounding line to retreat by several km inland during MIS-3, stopping at only 40 km far from the PD position. Grounding-line advance and retreat is often very rapid, especially during the first advance after the LIG or during the MIS-3. This is primarily due to the oceanic forcing applied, since the large advance and retreat well follow submarine melt rate evolution. Part of this stepped nature may be due to the bathymetry too. The area connecting ZI and 79N to the inland shows a bedrock depth of 100-200 m (Morlighem et al., 2017). In periods of relatively high sea level, such as the first kyr after the last interglacial, this deep bathymetry may be crucial in driving the grounding line evolution (in our model through the flotation criterion), since it hampers the ice to ground and so the ice sheet to advance. This is in line with recent work suggesting that deep bathymetry combined with warmer waters entering the fjord may have important consequences in the destabilisation of 79N (Schaffer et al., 2017).

The effect of submarine melt applied to the NEGIS marine margin during MIS-3 is also perceived far inland. The basal melt imposed at the ice-ocean interface causes the ice margin to retreat inland, strongly enhancing ice discharge (Fig. 5 and 6). The reduction of buttressing previously ensured by the presence of ice on the continental shelf increases margin velocities, which propagate inland (Fig. 4 f), causing a decrease of ice thickness in the ice-sheet interior (Fig. 6). An initial strong peak in ice discharge is observed, following the initial increase of submarine melting and loss of buttressing, but the effect persists with further ice discharge until the end of MIS-3. At this moment, the absence of melt imposed through the LGM allows the grounding line to advance again towards the continental shelf break (Fig. 4 g-j). The maximum distance reached at the peak of the LGM and the time of the onset of the advance are inversely proportional to the melt rate suffered in the previous millennia (Fig. 1). A strong melt rate imposed during MIS-3 leads to a delayed triggering and spatially-constrained grounding-line advance,

and vice versa.

By construction, submarine melt occurs again after 20 ka BP, when both atmospheric and oceanic temperatures increase, contributing to push the grounding line back towards the ice-sheet interior (Fig. 3). This retreat is also simulated in the unperturbed experiment, which demonstrates that the Holocene ice loss is driven by both increasing atmospheric and oceanic temperatures.
Nevertheless, the presence of submarine melt at the NEGIS marine margins enhances the retreat and triggers it slightly earlier. This feature, then, saturates for $B_{\mathrm{ref}} > 3\,\mathrm{m\,a^{-1}\,K^{-1}}$, as further inland retreat is constrained by the bathymetry. However, to specifically assess the relative role between atmospheric and oceanic forcings in the evolution of the NEGIS margin, an equal sensitivity test on the atmospheric forcing, and/or further experiments with another melt scheme, should be carried out.

## 4  Discussion

### 4.1  Comparison between modelled and data-derived reconstructions

The NEGIS grounding-line fluctuations simulated in response to a high oceanic forcing in this set of experiments are similar to those suggested by Larsen et al. (2018) for the last 45 ka. This reconstruction is a result of averaging the evolution of three NEGIS outlet glacier fronts (79N, ZI and SG) inferred from the various geological records with respect to their position at 2014 (Howat et al., 2014). Although it is a valuable tool providing a rough idea of the margin fluctuation during the last 45
ka, caution should be taken before performing a precise one-to-one comparison with model data. Specifically, while the strong retreat during the Holocene is documented for all those glaciers, records showing their margin position during MIS-3 are available only for ZI and SG, which were behind their present location by ca. 20 and 40 km, respectively. However, since they all shared the same behavior during the Holocene, it is likely that 79N front was as far inland as the others during MIS-3 (Larsen et al., 2018). Made this premise, there are some major differences between our results and theirs that deserve further attention.
First, we do not simulate the MIS-3 retreat farther inland than the PD position (20-40 km), although our simulations do show a retreat of more than 100 km with respect to the previous millennia. The observed MIS-3 retreat behind the PD NEGIS margin position has been attributed by Larsen et al. (2018) to lower accumulation rates, high incoming solar radiation and increasing summer air temperatures operating together. Since we have not investigated the sensitivity to these forcings separately and our experiments do not show this extended retreat, we can neither confirm nor discard their hypothesis. However, our work has
demonstrated that orbitally-driven oceanic warming during MIS-3 is enough to cause a substantial retreat of the NEGIS margin during part of the last glacial period.

Second, our simulated grounding-line advance during the LGM is smaller than the maximum extension suggested by reconstructions from geological records (Fig. 7). This bias furthermore increases with increasing oceanic forcing. Even in the unperturbed experiment, which allows the largest ice-sheet expansion due to the absence of melting at the marine margins,
the grounding line does not reach the continental shelf break. Nevertheless, our simulated extent is still one of the best reconstructions of northeast Greenland at the LGM obtained with an ice sheet model (Bradley et al., 2018; Lecavalier et al., 2014; Simpson et al., 2009; Tabone et al., 2018). This discrepancy in the LGM extents is reflected in the transient GrIS sea level contribution from the LGM to the present (Fig. S4), that is underestimated as compared to other recent modelling work (Lecavalier

et al., 2014; Tabone et al., 2018). Nevertheless, our estimation is not far from others (Huybrechts, 2002; Simpson et al., 2009) and well within the range proposed by Buizert et al. (2018). Note that although the LGM extent simulated by Lecavalier et al. (2014) is smaller than ours in the northeast, their ice volume contribution at the glacial maximum is about 1 m SLE higher. This could be partly due to their larger grounding-line advance in the northwest, but it might be also related to a more active dynamics in our simulations. This is plausible since also the volume discrepancy between our two studies, both performed using the same ice-sheet-shelf model, is likely due to differences in the dynamics. The main reason seems to be related to the fact that SIA and SSA velocities are here simply summed up instead of mixed through a weighting function as in Tabone et al. (2018). This increases the velocities in the transition zones, promoting discharge of ice from the interior and consequently limiting the ice volume accretion. Second, Tabone et al. (2018) accounted for refreezing processes at the base of the ice shelves, which allowed the grounding line to advance easily, leading to a glacial state in which almost all the GrIS margins were able to reach the shelf break. It is clear that this larger extent could account for a substantial part of the ice volume discrepancy. Another possible reason could be that here we increased the basal drag at the base of grounded temperate ice (by increasing its coefficient $c_{\mathrm{f}}$, of Eq. 2). More friction at the base may foster the production of water at the ice-bed interface through heat release, lubricating the bed and causing the ice flow to accelerate. However, we expect that this process is responsible for only a small fraction of the ice volume discrepancy, since it is counteracted by the increase in basal friction itself. Increasing the total ice volume during the glacial (and its extent) would probably require a substantial tuning effort, that is beyond the scope of this study. Our goal is not to provide a perfect match with the LGM but to illustrate a plausible mechanism behind the retreated ice margin at MIS-3 and its subsequent advance.

Third, the timings of the grounding-line advance/retreat for the last 45 ka of the last glacial period do not precisely correspond to those proposed by Larsen et al. (2018). In the experiments that show a significant retreat during MIS-3 ($\kappa > 4\,\mathrm{m\,a^{-1}\,K^{-1}}$), we simulate both the grounding-line advance at the end of this stage and the retreat at the onset of the Holocene earlier than expected. This is due to the submarine melting signal representing oceanic temperature anomalies, which saturates at zero at around 35 ka BP and is switched on again at 20 ka BP, assuming that the LGM starts and ends at these times. Since both atmospheric and oceanic forcings are built through the same index $\alpha$, any uncertainty in their evolution at orbital timescales affects the ice-sheet retreat during the Holocene which is supposed to be a combination of atmospheric and oceanic temperatures (Larsen et al., 2018). Although the Holocene maximum is quite well reproduced in our submarine melting configurations (Fig. 1) and in the atmospheric temperature evolution, the slight basal melt decrease applied in the late Holocene is not sufficient to make the grounding line advance back towards the continental borders and the inaccurate position simulated at the PD is a direct consequence of this simplification. Generally, the grounding-line retreat at the PD is proportional to the magnitude of the submarine melt rate imposed during the mid- and late Holocene, which is related to the value of $B_{\mathrm{ref}}$ used. However, this correspondence is very weak and the retreat quickly saturates at about 70 km away from the PD position along the glacier flow direction for $\kappa > 3\,\mathrm{m\,a^{-1}\,K^{-1}}$, where it is stopped by the presence of a bedrock above the sea level (Schaffer et al., 2016; Morlighem et al., 2017). Although such a retreat is supported by proxies for the mid-Holocene (Bennike and Weidick, 2001; Larsen et al., 2018), its persistence until the present day is clearly unrealistic. Moreover, it is likely that imposing PD submarine melting peaks of 50-75 $\mathrm{m\,a^{-1}}$ as estimated at the 79N grounding line (Anhaus et al., 2019; Wilson and Heimbach, 2017),

which are even higher than the $B_{\text{ref}}$ values considered in this work, could cause a further inland retreat. This bias could be related to 1) the low spatial resolution of the model (10 km), which does not allow for a precise treatment of the grounding-line zone and may trigger non-linear effects, enhancing grounding-line retreat farther inland than expected, 2) the design of the submarine melt signal itself during the Holocene, which shows a constant increase from 0 to the set $B_{\text{ref}}$ value through the last 20 ka. It is unlikely that such a monotonic increase could have persisted for most of the Holocene, since several records inferred from sediment cores in the Arctic Ocean and in Fram Strait indicate that water temperatures strongly fluctuated during the Holocene due to the variability of the oceanic currents. Surface (Sztybor and Rasmussen, 2017) and subsurface (Werner and Kandiano, 2013; Werner and Polyak, 2016) oceanic temperatures increased by 3-4 K since the beginning of the Holocene due to the inflow of Atlantic warming waters. After 9-8 kyr, however, these records report a drop in temperatures, gradually (Falardeau et al., 2018; Werner and Polyak, 2016) or interrupted by some peaks of warming (Consolaro et al., 2018). These different oceanic conditions between the early and mid- and late Holocene suggest that such a continuously high melting rate during the whole Holocene is likely overestimated.

## 4.2 Oceanic forcing at orbital timescales

The oceanic forcing is defined here to be in phase with the atmospheric forcing, as they are both set to evolve in time through the same NGRIP-derived index $\alpha$. To our knowledge, little evidence on oceanic changes at orbital timescales is available and whether the best representation of reality would be through oceanic temperatures varying in phase or antiphase with the atmosphere is unclear. However, proxy-based temperature reconstructions indicate glacial-interglacial surface temperature anomalies to be between 0 and -3 K (Annan and Hargreaves, 2013; MARGO, 2009) and the value chosen for $T_{\text{LGM,ocn}} - T_{\text{PD,ocn}}$ is within this range (-1 K).

The rapid occurrence of warm oceanic pulses on millennial timescales is an important characteristic of MIS-3, which is not taken into account here. Available records based on sediment cores of the Arctic Ocean suggest rapid temperature fluctuations as a result of large changes in water masses at different depths. Given the non-linear response of subglacial melting to temperature variations (e.g. Mikkelsen et al. (2018)), this effect could potentially modulate the orbitally-driven response on shorter timescales. Generally, strong oceanic variations are found between glacial-interglacial, but also between larger stadial-interstadial transitions (Poirier et al., 2012). High SST and low intermediate water temperatures are typical of interstadials, with warmer surface waters generally lasting for 3-4 kyr before cooling (Müller and Stein, 2014), while the opposite is found during stadials, when warmer Atlantic subsurface waters enter the Nordic Seas up to the Arctic Ocean (Rasmussen et al., 2014). Such a strong oceanic temperature variability is also documented by a stack of sediment cores of the Arctic Ocean and the Fram Strait for the last 50 kyr, suggesting that several peaks of warmings occurred during MIS-3 reaching temperatures 1-3 K higher than those recorded for the Holocene (Cronin et al., 2012). Nevertheless, the evolution of this temperature record at long (orbital) timescales agrees qualitatively well with that of the melting rate signal used in this work: high melting during MIS-3, prolonged cooling during the LGM and resumed melting during the Holocene. Thus, even though we remove some degree of realism by not considering the millennial-scale variability in the ocean, our experimental design could fairly well

represent the evolution of northern Greenland oceanic conditions over long timescales. A complete treatment of the problem from this perspective is difficult, however, by the lack of paleoceanographic records of the northeastern part of Greenland that provide information on the oceanic state during most of the last glacial period at high temporal resolution.

## 4.3 Model performance on the whole GrIS

The comparison of our results with observations is a good strategy to assess the model performance and to comprehensively evaluate the robustness of our results. At large spatial scales our simulations represent the present state of the GrIS reasonably well (Fig. S1). The maximum differences in surface elevations are found in the southwest and in the east due to a mismatch in ice cover. There, the ice sheet ends in many steep and narrow fjords which are not adequately represented by the 10km-resolution model. Also, the NEGIS front is located farther inland than observed. The velocity field shows a pretty good agreement in the interior of the ice sheet, where ice speeds are expected to be lower than $50 \, \mathrm{m \, a^{-1}}$ (Joughin et al., 2018). However, the simulated ice flow of outlet glaciers and ice streams shows more discrepancies. The speed of the inland flow is generally overestimated, whilst the velocities of streams as they extend far inland is underestimated. By zooming into our domain of interest we see that this pattern is also shared by the NEGIS (Fig. S2 and left panel of Fig. S3). The stream geometry is not adequately resolved, although the spatial distribution of the velocities is somewhat consistent with observations (faster flow at the margins and reduced speed in the interior, as seen in Fig. S2). However, the fast flow features of the tributary that feed the 79N are not reproduced; the SG is faster than expected and, instead of the long penetrating tongue of ice that characterises the NEGIS, the model simulates a stream drainig a wider area. Properly modelling the NEGIS is a well-known problem of ice-sheet models that investigate the evolution of the GrIS at large spatial scales. Most of these models underestimate the stream velocity and do not properly capture its outline (Aschwanden et al., 2016; Calov et al., 2018; Golledge and Edwards, 2019; Greve and Herzfeld, 2013; Seddik et al., 2012). Greve and Otsu (2007) succeed in reproducing a correct magnitude of its speed by increasing the basal sliding under the NEGIS by three orders of magnitude relative to the rest of the ice sheet, but they fail in reproducing its geometry. A good agreement between model and data is found in Price et al. (2011) and Peano et al. (2017), who use a spatially variable basal friction coefficient derived from an iterative inverse method to match the observed velocities. Our imperfect reproduction of the NEGIS is probably related to a combination of low spatial resolution (10 km) and problems in capturing the dynamics at the base of the ice sheet. Our basal friction coefficient $\beta$ is a function of the effective water pressure at the base of the ice sheet (Eq. 2), which is a significant degree of freedom in ice-sheet models. A better representation of basal hydrology and sliding could help to improve the simulation of the ice stream. In parallel, new studies on the origin of the stream (following Rogozhina et al. (2016)), its basal characteristics (following e.g. Keisling et al. (2014), Christianson et al. (2014) and Riverman et al. (2019)) and new data from the EGRIP ice core (following e.g. Vallelonga and Winstrup (2014)) will bring new insights in this direction.

Also, the model behaves satisfactory in simulating the advance and retreat of the GrIS margins throughout the last 120 kyr (see also Tabone et al. (2018)). Part of this performance is related to the two-condition calving law, that is a function of the critical ice thickness $H_{\mathrm{f}}$ below which the ice edge is calved. Thus, depending on its value, this law may be more or less conserva-

tive. Here, with an imposed threshold of 200 m, both 79N and ZI floating tongues are lost at the present. It could be that by increasing the value for $H_f$ the model would show slightly more resistance to calve. However, the impact of the calving law is limited to the grid cells at the ice front, while the retreat caused by the submarine melt involves fluctuations in the margin of hundreds of km. Alvarez-Solas et al. (2019) assessed this issue in a different context (the sensitivity of the Eurasian ice sheet to

the oceanic forcing during the last glacial period). Their results showed that the overall effect of this parameter is to modulate the amplitude of the response to the oceanic perturbations but its value did not qualitatively affect their main results. Thus, we expect that changes in the calving critical thickness may cause only second-order effects on the retreat.

## 4.4 Future perspectives

This work represents the first attempt to simulate the striking margin retreat reconstructed for the NEGIS during MIS-3 (Larsen et al., 2018) only accounting for changes in the oceanic forcing. However, such an ocean-driven retreat of the ice margin may have triggered feedbacks on the local climate that are not taken into account in this work. For example, it is possible that this large ice retreat would have caused changes in the albedo, affecting surface air temperatures and snow accumulation. Other feedbacks related to the freshwater flux into the ocean could have led to variations in sea ice and local oceanic circulation.

All these processes, not included here, could have additionally contributed to variations in the ice thickness and grounding-line position, and should be investigated in the future for a complete understanding of the conundrum. Further experiments accounting for changes in the atmospheric temperatures and precipitation or variations in the external forcing (i.e. insolation) should be carried out for a full understanding of the mechanisms involved in this retreat, here explained by considering the sole impact of the ocean. Particularly, a sensitivity study on climatic variations performed with a prescribed ocean could help

constraining the effect of the atmosphere to eventually evaluate the relative role between the forcings in driving the NEGIS margin.

## 5   Conclusions

We have studied the sensitivity of the NEGIS ice margin to oceanic forcing during the last glacial period. To this end, we used a three-dimensional, hybrid ice-sheet-shelf model in which the submarine melt rate is parameterised to perform simulations of

the GrIS for which basal melt follows a ice-core-proxy-derived curve assumed to represent the evolution of both atmospheric and oceanic temperatures at orbital scales. The increase in basal melt during MIS-3 reflects a relatively warm oceanic state, whereas the lack of basal melt during the LGM corresponds to the associated expected minimum in oceanic temperatures. We showed that in the absence of submarine melting during the entire last glacial period, the grounding line advances towards the continental shelf just after the LIG. On the other hand, switching on the oceanic forcing helps to limit the ice margin

advance. Specifically, sufficiently high submarine melt rates during MIS-3 eventually trigger its retreat by more than 100 km from its former position. The lack of basal melt during the LGM then resumes the grounding-line advance by 200 km towards the continental shelf break. Our results robustly show that a prolonged presence of submarine melt at the NEGIS ice margin

is enough to substantially contribute to grounding-line retreat there, which helps to explain the recently suggested NEGIS ice margin retreat during MIS-3.

*Author contributions.* IT performed the experiment, analysed the results and wrote the manuscript. All the contributors of this work helped to conceptualize the experiment and write the paper.

5 *Competing interests.* The authors declare that they have no conflict of interest.

*Acknowledgements.* We acknowledge the Spanish Ministry of Science and Innovation for supporting this work in the framework of the projects MOCCA (Modelling Abrupt Climate Change, grant no. CGL2014-59384-R) and RIMA (Reducing uncertainty in the evolution of ice sheets, grant no. CGL2017-85975-R). IT was funded by the Spanish National Programme for the Promotion of Talent and Its Employability (grant no. BES-2015-074097). AR was funded by the Ramón y Cajal Programme of the Spanish Ministry for Science, Innovation and 10 Universities. The model simulations were performed in the HPC of Climate Change of the International Campus of Excellence of Moncloa (EOLO), supported by MECD and MICINN. We are grateful to two anonymous referees, Kerim Nisancioglu and the handling editor Andreas Vieli for their valuable help in improving this work. Also, we are thankful to Catherine Ritz for providing the original model GRISLI.

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

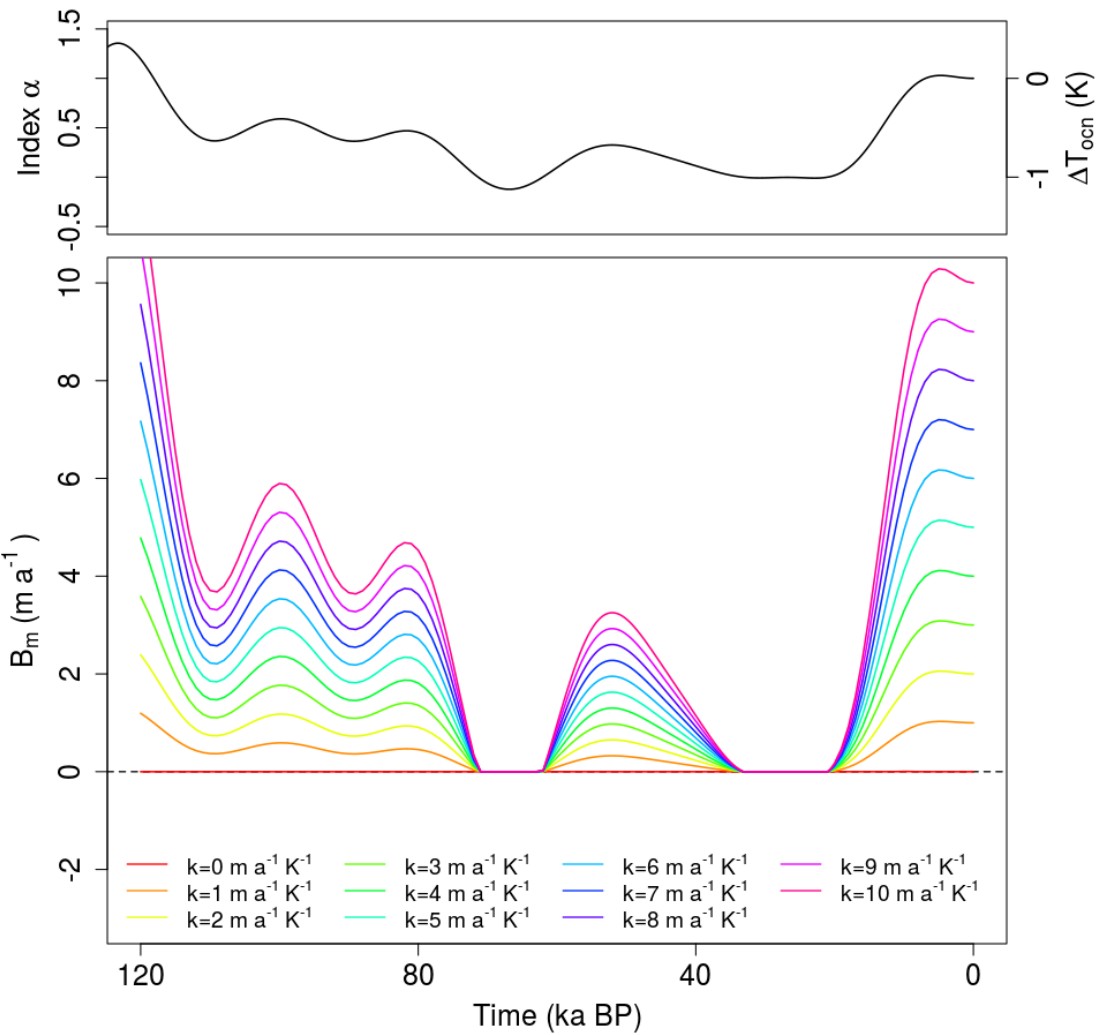

**Figure 1.** Evolution of the climatic index $\alpha$ and the resulting past oceanic temperature anomaly $\Delta T_{\text{ocn}}$ (K) (upper panel). Potential submarine melt-rate evolution during the last glacial period for increasing $B_{\text{ref}}$ and $\kappa$ values considered in the experiments (lower panel).

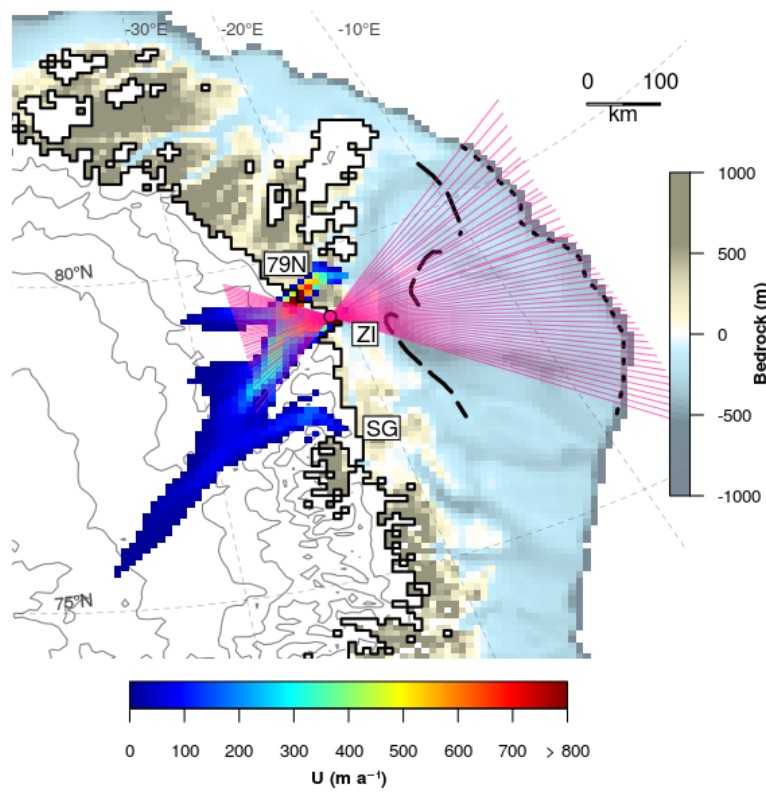

**Figure 2.** Map of the NEGIS sector showing the location of its three outlet glaciers (79N, ZI and SG), the observed present-day grounding-line position (solid black line), the observed present-day surface velocities (from Joughin et al. (2018)), the offshore bathymetry and the onshore ice elevation (both from Schaffer et al. (2016)) and the maximum (dotted black line) and minimum (dashed black line) grounding-line positions reconstructed for the LGM (Funder et al., 2011). The 48 transects used to calculate the evolution of the grounding-line position are shown in purple.

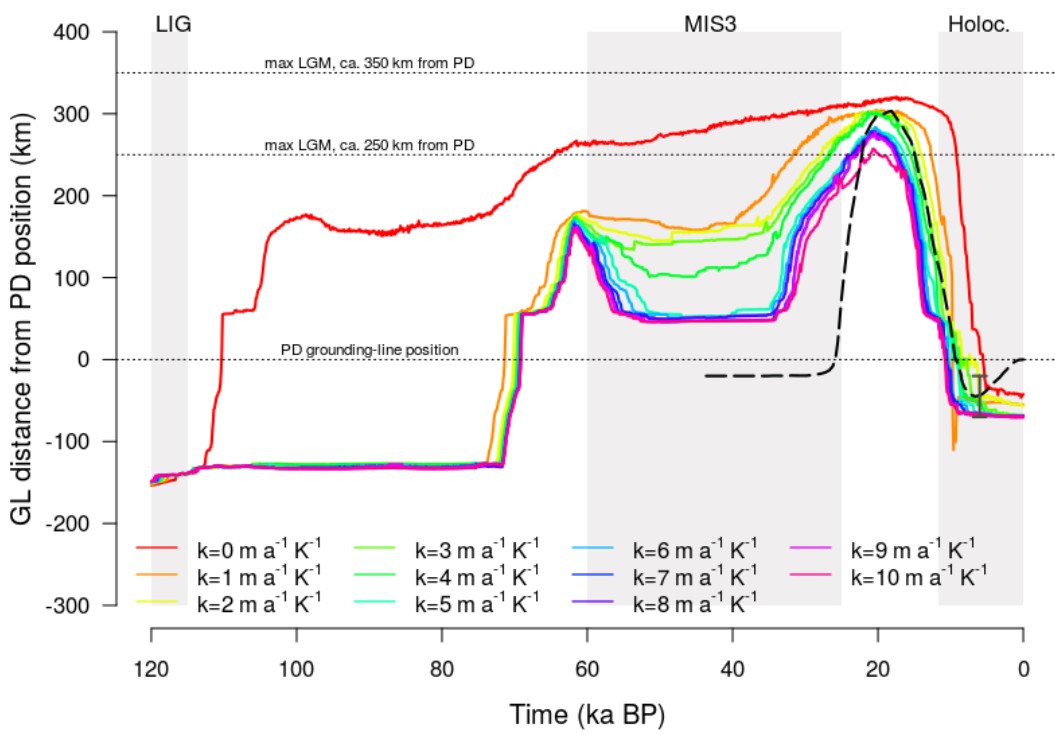

**Figure 3.** Simulated evolution of the NEGIS grounding line relative to its observed present-day position for the set of experiments (coloured lines). The grounding-line distance has been calculated along 48 transects which follow approximately the flow direction of NEGIS ZI glacier towards the shelf break (Fig. 2). The dashed black line shows the reconstruction by Larsen et al. (2018). Shaded regions represent the time periods corresponding to the LIG, MIS-3 and the Holocene. The three dotted lines show the PD NEGIS grounding-line position (0 km) and the maximum (300 km $\pm$ 50 km) expected advance of the northeast GrIS to the continental shelf break at the LGM according to Funder et al. (2011).

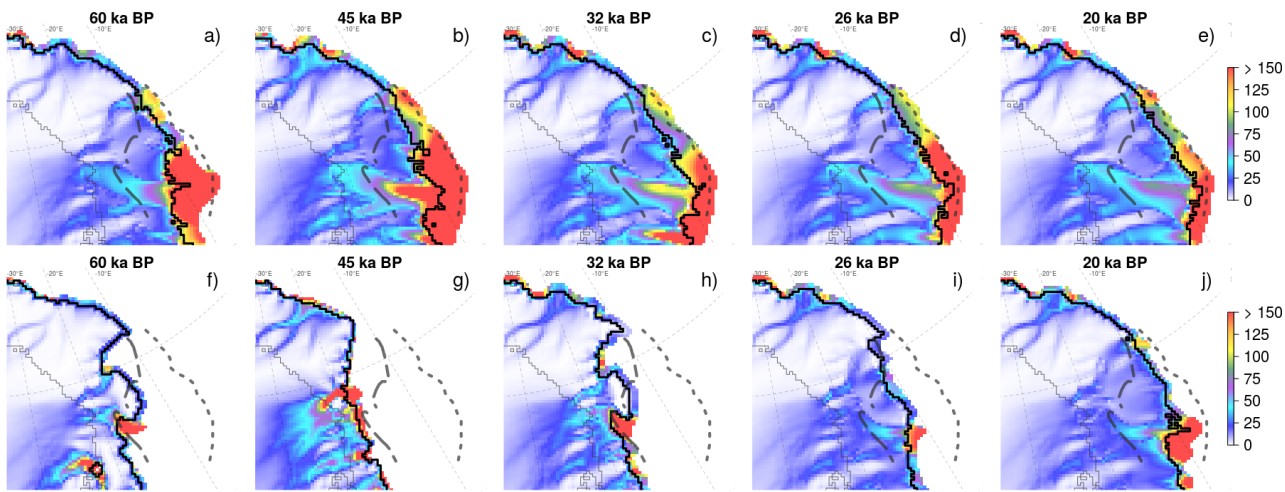

**Figure 4.** Snapshots of U ($\text{m a}^{-1}$) in total absence of submarine melting (a-e) and in presence of active orbital-driven oceanic forcing ($\kappa = 8 \text{ m a}^{-1} \text{ K}^{-1}$, $B_{\text{ref}} = 8 \text{ m a}^{-1}$) (f-j) at different times along MIS-3 and the LGM. The shown sector spans an area of about 600 km by 600 km. The black line represents the position of the simulated grounding line. The grey thin solid line represents the observed PD grounding-line position (Schaffer et al., 2016). Maximum (dotted black line) and minimum (dashed black line) grounding-line positions reconstructed for the LGM (Funder et al., 2011) are also shown.

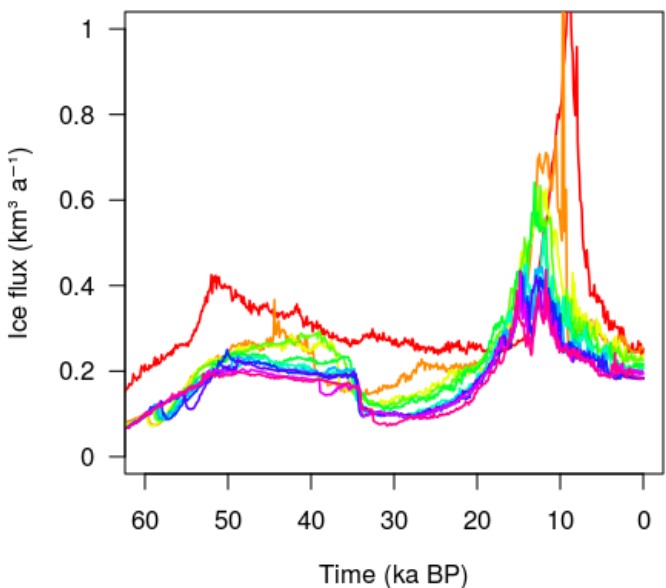

**Figure 5.** Simulated ice flux at the NEGIS sector for different oceanic forcings. Colors refer to the color scale of Fig. 3.

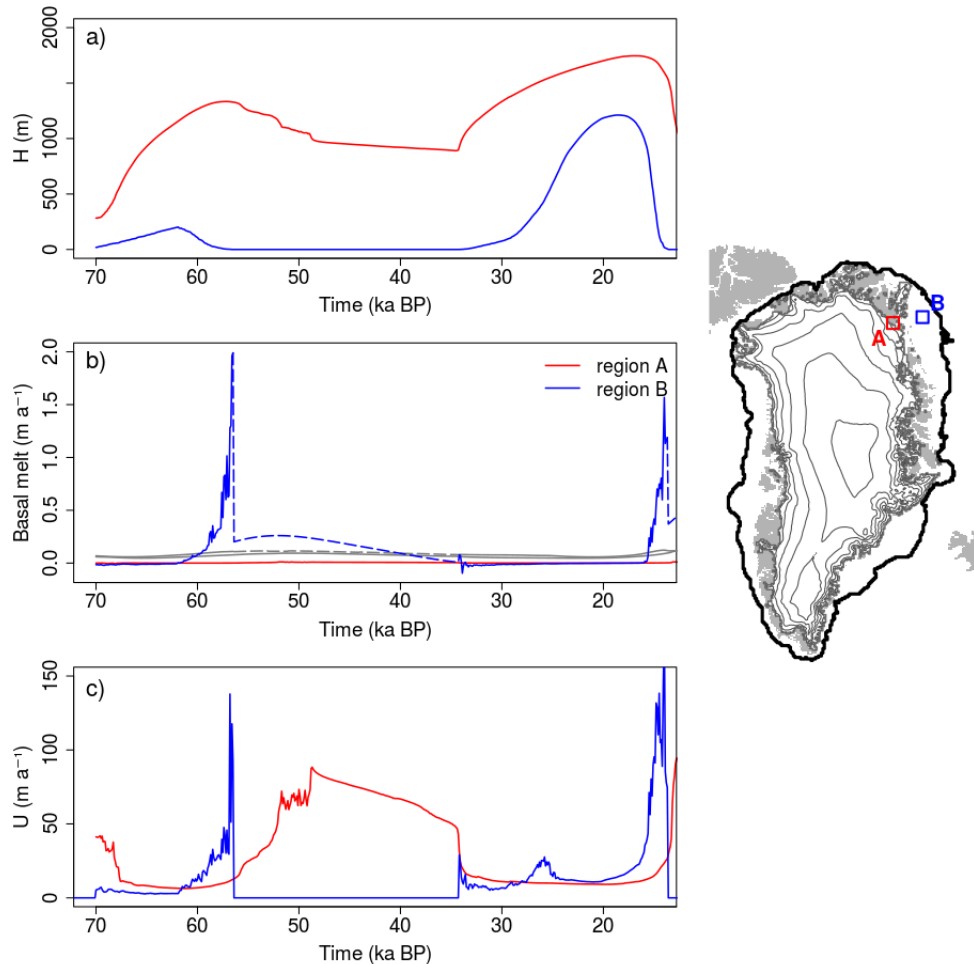

**Figure 6.** Evolution of the ice thickness (a), basal melt (b) and ice velocity (c) averaged within the regions A (red lines) and B (blue lines), simulated in presence of submarine melt during MIS-3 ($\kappa = 8 \ \mathrm{m\,a^{-1}\,K^{-1}}$ experiment). Grey lines in panel b) show the contribution to surface mass balance (accumulation minus ablation) simulated by the model and averaged over the regions A and B. Dashed lines in the same panel show the potential contributions that would be observed if regions A and B were ice covered. The black line on the side map represents the LGM maximum extent for the $\kappa = 8 \ \mathrm{m\,a^{-1}\,K^{-1}}$ experiment.

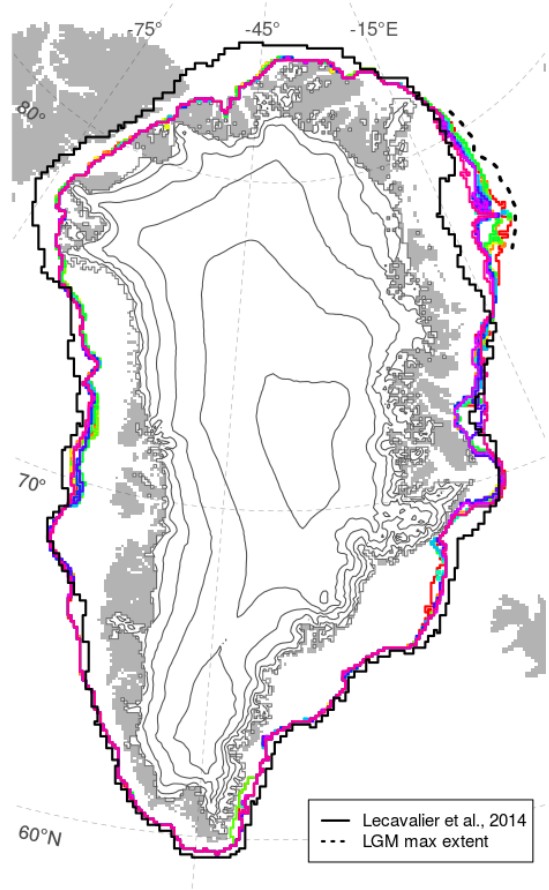

**Figure 7.** Simulated GrIS extent at the LGM for different oceanic forcings compared to other glacial reconstructions. Colored lines follow the color scale of Fig. 3. The solid black line refers to the maximum glacial extent simulated by Lecavalier et al. (2014), calibrated to match the minimum LGM configuration (Funder et al., 2011) in the northeast. The dashed black line represents the expected maximum glacial extent at the northeast sector as inferred from various geological data (Arndt et al., 2015, 2017; Evans et al., 2009; Winkelmann et al., 2010).