# Peer review of "Submarine melt as a potential trigger of the North East Greenland Ice Stream margin retreat during Marine Isotope Stage 3."

_The Cryosphere, 2018_

## Referee Comment (RC1) · Anonymous Referee #1 · 21 Dec 2018

This paper addresses an interesting question about the past behaviour of the NEGIS system in Greenland. Its is relevant to the readership of TC and is a useful investigation of paleo-retreat controls in this area. The conclusions reached are reasonable based on the relatively simple experimental design (although see problem below). The work is linked to the appropriate existing literature in the region, and is mostly well written. In a few places the language used needs rephrased (see below) for clarity.

The main problematic issue I found with this paper is that it is very unclear whether the advance condition fits well with the geological evidence for LGM extent and therefore it is further unclear whether the retreat is to any degree preconditioned by this lack of fit. Thus the magnitudes and rates of retreat are harder to trust when we don't see the fit between the model and the data in a clear way. This could be solved by

[Figure]

a more rigourous discussion of fit, and with figures enabling the fit to be visualised and quantified. That said, most of the sensitivity tests show advance to a fairly similar distance from the GL and therefore the comparative nature of the sensitivity tests is useful. In addition, the fact that sustained melt could have driven the retreat seems like a conclusion that wouldn't change if there were a different fit with the model extents and geological data.

Overall, with this issue addressed above, and the general comments addressed below, then this should be a useful contribution to the understanding of the NEGIS system.

General Comments:

Page 1:

L5 'important conundrum' – please indicate more clearly what the conundrum actually is. I.e. you outlne that the ice stream is losing mass and retreating over the last decades and that it retreated further inland at MIS3. Why is this a conundrum?

L6: 'a modelling approach is pending' – you mean a modelling approach has never been used to test the hypotheses?

L14: I would like to see a good location map with the detail of the glaciers (and names) picked out and the overall setting shown clearly.

L16: 'almost lost' – can this actually be quantitatively described because otherwise this is quite a vague statement.

L17: 'due to its bed configuration' – please describe the differences in bed configuration – add detail and explain what you mean.

L18: 'lost mass' – can the mode of mass loss be described? E.g. thinning? Retreat? Both?

Page 2:

L5: delete 'even'.

L10: delete 'further back in time'. Also, what is meant by 'largely'? E.g. the fluctuations were large in magnitude? Or most of the ice margin fluctuated?

L11: delete 'even'.

L10-12: You don't mention that there must have been readvance since MIS3. Perhaps indicate what is known about this too?

L12-13: 'The Holocene retreat.... Etc.' – this sentence conflates the past and the future. It would be good to separate out the past bit and then say why it is important to look at this - e.g. because conditions may have been similar to what is expected in the future. In other words, this is the justification for why the paper is a useful piece of work.

L13-14: delete 'On the other hand' as it is superfluous text.

L15: 'was undertaken yet' should be 'has yet been undertaken'.

L30-31: The frictin law is mentioned, but not quantitatively described. Can it be described more quantitatively?

Page 3:

L1: Can a comment be made about whether the flotation criterion and calving model has implications for any particular behavioural characteristics of the model. E.g. will this still produce good overall responses in terms of space? Will the rates of retreat (or readvance) be expected to be robust (or too fast or too slow?). Does it deal with retreat vs. readvance hysteresis well? Perhaps a general paragraph on what we know this model is good at in general would therefore be useful.

L14: I was surprised to see that there was no sensitivity testing of the climatic controls on the model experiments. Can you comment somewhere in the text as to whether you think the results would be significantly different if the climate control was altered within

a certain range of uncertainty?

Page 4:

L18: Can you justify why basal melt rate is not parameterised in a depth-dependent manner? Or in other words, why is a 10% Bm on all floating grid cells an appropriate decision to make?

L25 or thereabouts: Do you make any assessment of whether the pre MIS3 state is realistic? I.e. how good is the spinup, can it be assessed, and how does it fit to the geological/field data from the region. In addition, can you confirm you are you allowing the grounding line to evolve through time? And does the ice shelf characteristics evolve through time furing spinup?

L30: I think a map of the fit between the pre MIS-3 state and the geological evidence would be an important figure to show. This will allow better discussion of whether the system is appropriately setup for the retreat experiments. I.e. if the extent or thickness is not correct, then how can we trust the degree or rates of retreat?

L32: delete 'already'.

Page 5:

L1: What is meant by 'substantially steady' – describe the margin stability pattern in clearer detail.

L21: 'stationing' isn't a good word to use. Do you mean 'stopping', or 'retreating to'?

L27: you mention that there is no melt imposed at the LGM. You could discuss somewhere later about whether you think this is a realistic condition.

Page 6:

L1 'saturates for high values' – can the high values be stated quantitatively?

L14-19: As mentioned before – I would like to see a better exploration, and a figure,

showing the fit of the LGM expansion and the field data. I think the weakness of this paper lies in both a lack of description of this, but also the fact that the fit is not as good as it could be. We really need to see how good the fit is so we can better judge the results. In addition, were there no modifications, for example to the climate, or the the Bm during the advance phase, that would help enable a better fit to the data? Some more sensitivity tests on this would have been good to see.

L17: 'insufficient basal drag' . Please tell us why the imperfect drag imposition would alter the result in this way?

Page 7:

L1: can you describe the pattern of saturation in more quantitative detail?

L9: 'it is unlikely that this could have happened for a long period of time and in such a persistent way.' Can you justify why this is the case? Explain in more detail – link to any knowledge in ocean circulation change etc.

Page 8:

L4: 'helps to constrain' – so you mean 'helps to limit'? Constrain could be interpreted in a number of ways – e.g. to limit or to provide evidence to help understand.

L7: It would be useful to know whether the 'prolonged presence of submarine melt' is something that is a realistic prospect based on any other evidence.

Figures:

Fig1: I would separate the inset map to a separate new figure 1 which should be a location map showing the NEGIS area in much more detail, including the key outlets, the location of the profiles, the offshore bathymetry, the onshore ice cover etc. In addition, either on that map or on an additional new figure, all the evidence for past ice extent should be shown so that we can then use it to judge how well the model fits with the geological dataset.

In addition, in this figure the growth and retreat steps are rapid. Can you say much about whether these are purely a function of the forcing provided, or whether the bed topography or fjord width is having any particular control on the 'stepped' nature of advance or retreat?

Given that you mention that there was a retreat to a position inland of the present day grounding line position, can you also show a horizontal dotted line to represent the knowledge of where this inland retreat reached?

Fig3: These figures are a little hard to follow because of their size. Can they be made bigger? The arrows pointing to the PG grounding line positions aren't terribly useful – surely a line on the map would be more appropriate. Finally, you mention these are snapshots at different times along MIS3 and the LGM. Please state which times these actually represent.

---

## Referee Comment (RC2) · Anonymous Referee #2 · 2 Jan 2019

The objective of the study is to test whether submarine melt (ocean warming) could be the primary cause of the ice margin retreat of NEGIS during MIS3 and MIS1 that was recently documented by Larsen et al 2018 using radiocarbon dating of reworked shells in historical (LIA) moraines. It uses the GRISLI-UCM 3D ice-sheet-shelf model to simulate the influence of submarine melt using a variable amount of melt rates.

I am not an expert in ice sheet modelling and cannot evaluate if the model set-up is state-of-the-art, but the description of the model set-up is easy to follow and understandable. It also seems to be realistic melt-rates that have been used to force the model. The manuscript is generally well-written, and the model-data comparison provides new and interesting knowledge about the potential effect of ocean warming and submarine melt on the evolution of NEGIS. However, there are a few places where

minor revision is warranted. These are listed below.

Page 1 Title: I am not aware of the TC politics on using abbreviations in the title, but I would avoid using them. The title could be changed to: Submarine melt as a potential trigger of ice margin retreat of the Northeast Greenland Ice Stream during Marine Isotope Stage 3

L1: remove "area"

L5: Why is this a conundrum? – this should be explained in more detail.

L9: MIS3 = Marine Isotope Stage 3

Page 2 L11-: change to.. ….even retreating 70 km behind its present-day position from 7.8-1.2 ka during most of the mid- and late Holocene and 20-40 km from 41-26 ka during Marine Isotope Stage 3 (MIS-3, c. 60-25 ka).

L12: Stage NOT state

L20: change to (LIG, c. 128-116 ka)

Page 5 L15: change to (c. 116 ka)

Page 6 L20 and L24: I guess it should be the last 45 ka?

L34: change to mid- and late Holocene

Figure 1: I would make the inset map bigger and outline NEGIS – maybe as a panel next to the diagram. It would also be useful if the LIG, MIS3, LGM, Holocene time periods are as shown as vertical bars.

Figure 3: I would suggest making the figure bigger as it is difficult to see the details in the maps. Maybe an outline of NEGIS could be placed on top of the velocity fields? It would also be valuable for the discussion if the LGM reconstruction of Funder et al and the minimum reconstructions (MIS-3 and MIS1) of Larsen et al could be shown on the maps.

[Figure]

---

## Referee Comment (RC3) · Anonymous Referee #3 · 24 Jan 2019

The study of Tabone et al. focuses on the Northeast Greenland Ice Stream (NEGIS) and its response to changes in climate, and in particular submarine melt, during the last glacial period. By applying climate forcing mimicking conditions during the last glacial period, an ice sheet/ice shelf model is used to study the transient evolution of the Greenland ice sheet over the past 120ka years. The evolution of the NEGIS are discussed in light of existing reconstructions of its history.

The study is original in assessing the long term response of the NEGIS to changes in climate, and goes beyond state of the art by comparing the dynamical evolution of the ice stream to proxy records. The paper is well written and the figures are clear. However, there are a several concerns which should be considered before publication in the cryosphere.

[Figure]

GENERAL COMMENTS: The results of the study are clearly novel and of great potential in our understanding of the long term evolution of the NEGIS. However, there is a lack of detail in the description of the model results and the full potential of the study is untapped.

Given that the model simulates the entire Greenland Ice sheet these results should be included and discussed. In particular, how well does the model reproduce the present data ice sheet configuration as well as the ice stream. Similarly, how do the model results compare to published simulations and reconstructions of the LGM configuration of Greenland. This should also include an assessment of the transient evolution of the equivalent sea level contribution from Greenland.

For the NEGIS it is not clear how well the ice stream itself is reproduced by the model. To what extent does the model capture the observed geometry and velocities of the ice stream? And in particular, an assessment of the time evolution of the ice stream should be included. In what periods was the ice stream active, and did it change its position through time? If possible the model simulations should be compared with reconstructions from marine sediment archives. To make these comparisons relevant, as more data on the evolution of NEGIS become available, a time series showing the simulated ice flux at the margin of NEGIS should be included.

Another concern is the choice of oceanic forcing applied to the model ice sheet. For simplicity the submarine melt rate is assumed to be spatially uniform around Greenland. Given the lack of data this can be argued to be a fair assumption. However, the impact of this choice should be documented and discussed in light of existing data from sites along the margins of Greenland. A bigger concern is the inference that past oceanic temperatures below the ice evolve in phase with the atmospheric temperature (eq. 4). Several studies have shown that during glacial periods the subsurface temperatures off Greenland were relatively warm due to the stratification of the water column under an extensive sea ice cover and associated fresh surface layer (see e.g. Alvarez-Solas et al. 2011).

[Figure]

SPECIFIC COMMENTS:

Line 7, page 1: LGP is not a common acronym. Better to spell out last glacial period and if necessary use common acronyms such as the LGM to specify a specific period within the glacial period where appropriate.

Line 14, page 1: NG - a more common acronym for the 79N glacier in the literature is 79N.

Line 16, page 1: it is stated that 79N is more stable than ZI due to its bed configuration - please elaborate on this.

Line 5, page 2: the slow retreat of 79N suggested by Choi et al. is described as conservative. Why? Please elaborate.

Line 11, page 2: the ice is thought to have retreated 20-40km being its PD position during MIS3. How is this now? Please elaborate and include an assessment of the uncertainties.

Line 24, page 2: resolve "?"

Line 8, page 3: the climate forcing is composed of 3 different ice core reconstructions (Vinther, NGRIP, and NEEM). Substantiate why this is done, instead of using only one ice core record such as NEEM.

Line 9, page: why is the variability below orbital removed? What is the purpose of this? What is the model result given the full variability represented by the climate reconstructions? Is there a reason to believe the millennial scale variability should be neglected in forcing the ice sheet?

Line 11, page 3 and eq. 1: PD is referred to as interglacial. Please be more precise on definition of PD: interglacial, Holocene or present day?

Line 13, page 3: why use CLIMBER-3a and not PMIP for the LGM - interglacial climate? What is the impact of the choice of model?

Line 15, eq. 2, page 3: What is the rationale behind choosing the same approach for calculating precipitation as for temperature? Is this appropriate? What is the impact of this choice, please document. Note that P_LGM and P_PD are not described. How are these calculated?

Line 16, page 3: why use PDD and not scale SMB from MAR which is used for the temperature?

Line 17, page 3: it is claimed that using PDD does not "jeopardise" results as focus is no oceanic forcing. Note that this invalidates any comparison of the relative importance of atmospheric and oceanic forcing. Please elaborate on this point + check manuscript for consistency with the discussion of the importance of the oceanic forcing given that its relative role cannot be assessed.

Line 8, page 4: resolve "?"

Figure 3: what is shown here. Please specify time periods for each subplot.

Figure 4: Show A and B in relation to ice margin (e.g. in figure similar to 3). Specify where smb and Bmelt are taken from.

---

## Author Comment (AC1) · 9 Apr 2019

This paper addresses an interesting question about the past behaviour of the NEGIS system in Greenland. Its is relevant to the readership of TC and is a useful investigation of paleo-retreat controls in this area. The conclusions reached are reasonable based on the relatively simple experimental design (although see problem below). The work is linked to the appropriate existing literature in the region, and is mostly well written. In a few places the language used needs rephrased (see below) for clarity.

The main problematic issue I found with this paper is that it is very unclear whether the advance condition fits well with the geological evidence for LGM extent and therefore it is further unclear whether the retreat is to any degree preconditioned by this lack of fit. Thus the magnitudes and rates of retreat are harder to trust when we don't see the fit between the model and the data in a clear way. This could be solved by a more rigorous discussion of fit, and with figures enabling the fit to be visualised and quantified. That said, most of the sensitivity tests show advance to a fairly similar distance from the GL and therefore the comparative nature of the sensitivity tests is useful. In addition, the fact that sustained melt could have driven the retreat seems like a conclusion that wouldn't change if there were a different fit with the model extents and geological data.

Overall, with this issue addressed above, and the general comments addressed below, then this should be a useful contribution to the understanding of the NEGIS system.

We thank the reviewer for their valuable comments and suggestions. We agree that a clear description of how our model is able to reproduce the expected LGM extent was missing in the old version of the manuscript (MS). This concern has been principally addressed by adding a figure that clearly shows the fit between our simulations and the data (new Fig. 7). Also, we improved the description of how our ice-sheet-shelf model simulates the paleo and present state of the GrIS. Knowing the general behavior of the model and its performance in comparison with reconstructions will help to evaluate our results.

An exhaustive point-by-point answer to the comments is shown below.

Note that some changes we made in the experimental setup (described in this document) may have quantitatively affected the evolution of the grounding-line position throughout the last glacial. Thus, some details in the description of the results may have changed. Please refer to the figures of the new version of the MS when reading the answers below.

General Comments:

Page 1:

L5 'important conundrum' – please indicate more clearly what the conundrum actually is. I.e. you outline that the ice stream is losing mass and retreating over the last decades and that it retreated further inland at MIS3. Why is this a conundrum?

L6: 'a modelling approach is pending' – you mean a modelling approach has never been used to test the hypotheses?

The two points above imply that this paragraph needs clarification. We changed it to: "*Alongside, a recent study suggests that the NEGIS grounding line was 20-40 km behind its present-day location for 15 ka during Marine Isotope Stage (MIS) 3. This is in contrast with Greenland temperature records indicating cold atmospheric conditions at that time, expected to favor ice-sheet expansion. To explain*

*this anomalous retreat a combination of atmospheric and external forcings has been invoked. However, the ocean was not brought into play. Here we investigate the sensitivity of the NEGIS to the oceanic forcing during the Last Glacial Period (LGP) using a three-dimensional hybrid ice-sheet-shelf model. We find that a sufficiently high oceanic forcing could account for a NEGIS ice-margin retreat of several tens of km, potentially explaining the recently proposed NEGIS grounding-line retreat during MIS-3.”*

L14: I would like to see a good location map with the detail of the glaciers (and names) picked out and the overall setting shown clearly.

We added a new figure zoomed on the analysed sector showing the location and names of the outlet glaciers, the observed present grounding-line position, the LGM reconstructed grounding-line positions (max and min), PD observed surface velocities, offshore bathymetry and the onshore ice cover (new Fig. 2), as suggested by the reviewer herein and below.

L16: 'almost lost' – can this actually be quantitatively described because otherwise this is quite a vague statement.

This sentence has been changed to: *“In less than 15 years the ZI floating tongue has lost the 95% of its size as a result of an enhanced mass loss (Mouginot et al., 2015).”*

L17: 'due to its bed configuration' – please describe the differences in bed configuration – add detail and explain what you mean.

L18: 'lost mass' – can the mode of mass loss be described? E.g. thinning? Retreat? Both?

The two points above have been addressed by changing this sentence to: *“In less than 15 years the ZI floating tongue has lost the 95% of its size as a result of an enhanced mass loss (Mouginot et al., 2015). Concurrently, since 1999 the 79N ice shelf has lost the 30% of its thickness at the grounding line (Mouginot et al., 2015), contributing to its inland retreat by 2 km (Mayer et al., 2018). However, since 79N is retreating over an upward-sloping bed (Mouginot et al., 2015), it may be less prone than ZI to an unstable retreat. This has been recently examined through an ice-flow model pointing out that its floating tongue has to lose several tens of km of ice before the glacier becomes unstable (Rathmann et al., 2017). A larger stability of 79N has been recently tested under various future warming scenarios by another modelling study (Choi et al., 2017), suggesting that it may be related to the presence of pinning points (such as ice rises) near the calving front. “*

Page 2:

L5: delete 'even'.

Done.

L10: delete 'further back in time'. Also, what is meant by 'largely'? E.g. the fluctuations were large in magnitude? Or most of the ice margin fluctuated?

This sentence has been changed to: *“The paleo records emerging from this study, combined with a collection of geological data assembled in the last 20 years (Weidick et al., 1996; Bennike and Weidick, 2001; Evans et al, 2009; Winkelmann et al., 2010; Arndt et al., 2015, 2017), suggest that the ice margin considerably fluctuated in magnitude throughout this period.”*

L11: delete 'even'.

This sentence has been modified to answer the next two points raised by the reviewer.

L10-12: You don't mention that there must have been readvance since MIS3. Perhaps indicate what is known about this too?

L12-13: 'The Holocene retreat. . .. Etc.' – this sentence conflates the past and the future. It would be good to separate out the past bit and then say why it is important to look at this - e.g. because conditions may have been similar to what is expected in the future. In other words, this is the justification for why the paper is a useful piece of work.

The two points raised above indicate that this paragraph needs clarification. Thus, it has been changed to: "*Although the age of these LGM reconstructions is still poorly constrained, the combination of cosmogenic exposure and radiocarbon dating has recently facilitated the reconstruction of the position of the NEGIS over the last 45 ka (Larsen et al., 2018). The paleo records emerging from this study, combined with a collection of geological data assembled in the last 20 years (Weidick et al., 1996; Bennike and Weidick, 2001; Evans et al, 2009; Winkelmann et al., 2010; Arndt et al., 2015, 2017), suggest that the ice margin considerably fluctuated in magnitude throughout this period. Around 41-26 ka BP during Marine Isotope Stage 3 (MIS-3, c. 60-25 ka) the NEGIS front was ca. 20-40 km farther inland than today, then advanced by more than 250 km toward the shelf break at the Last Glacial Maximum (LGM) and retreated again during the last deglaciation, at ca. 70 km behind its present-day position, where it stopped most of the mid-and late Holocene (7.8-1.2 ka BP). The Holocene retreat was likely due to an increase in both atmospheric and oceanic temperatures, whilst the retreat during MIS-3 was attributed to a combination of atmospheric and external forcings. However, the potential role of oceanic forcing in this retreat has not been explicitly investigated from a modelling perspective. In the light of the ongoing changes in the GrIS attributed to ice-ocean interactions, this appears as a plausible mechanism that needs to be investigated. Moreover, since it is expected that warmer Atlantic waters entering the fjords will strongly affect the NEGIS margin in the future, assessing its response to similar past warm oceanic conditions will provide new insights into the future stability of its glaciers front.*"

L13-14: delete 'On the other hand' as it is superfluous text.

Done (see the paragraph above).

L15: 'was undertaken yet' should be 'has yet been undertaken'.

This sentence has been modified (see the paragraph above).

L30-31: The friction law is mentioned, but not quantitatively described. Can it be described more quantitatively?

We agree this aspect was lacking in the old version of the MS, thus we added a more detailed description of our friction law.

This paragraph has been changed to:

"*The SSA boundary condition is provided by basal sliding below the ice streams following a linear friction law, in which the basal shear stress $\tau_b$ is proportional to the basal velocity $u_b$ and to a friction coefficient $\beta$ dependent on the effective pressure of the water at the base of the ice sheet $N_{eff}$, as:*

$$\tau_b = - \beta * u_b \text{ (Eq.1)}$$
*where*

$$\beta = c_f * N_{eff}. \text{ (Eq. 2)}$$

*The term $c_f$ depends on the different features of the bedrock topography (e.g. presence of sediments); $N_{eff}$ is calculated as $N_{eff} = \rho g H – p_w$, where $\rho$ is the ice density, g the gravitational acceleration and H the ice thickness. The sub-glacial water pressure $p_w$ comes from a simple basal hydrological model based on a Darcy-type law, for which water flows at the base of temperate ice as driven by a gradient of hydraulic pressure. Despite the simplicity of this hydrology scheme, it provides a fair description of the outflow systems at the base of the ice sheet (Peyaud et al., 2007).*"

Page 3:

L1: Can a comment be made about whether the flotation criterion and calving model has implications for any particular behavioural characteristics of the model. E.g. will this still produce good overall responses in terms of space? Will the rates of retreat (or readvance) be expected to be robust (or too fast or too slow?). Does it deal with retreat vs. readvance hysteresis well?

Since other calving and grounding-line treatments other than the flotation criterion and the two-conditions calving law have not been implemented in our model, it is difficult to judge the behaviour of these laws compared to others. However we can make some considerations.

The same calving law has already been tested in previous work (e.g. Peyaud et al., 2007, Colleoni et al., 2014, Alvarez et al., 2018) and appears to reasonably reproduce the advance and retreat of the ice sheet throughout the time. It is suitable for hybrid coarse-resolution ice-sheet models used with paleo purposes since it is able to describe the calving processes in a simple way without demanding too high computational effort. Its satisfactory performance can be seen for example in Tabone et al., 2018, where with the same treatment of the margin front, it is shown how well the model is able to simulate the advance and retreat throughout the last two glacial cycles. However, this calving law is a function of the imposed critical ice thickness below which the ice block is calved and depending on its value, it may be more or less conservative. Here, we imposed a threshold of 200 m, and both 79N and ZI floating tongues are lost during the Holocene. This is also documented in Peano et al. (2017) for the present state of the ice sheet. It could be that by increasing the ice thickness threshold (below which the ice block is calved) in the calving law the model would show slightly more resistance to calve. Although this cannot be completely assessed without a sensitivity test on that parameter, it is likely that an increase in this critical thickness would allow for longer and rather more stable ice shelves. However, the impact of the calving is limited to the grid cells at the ice front, while the retreat caused by the submarine melt involves fluctuation in the margin of hundreds of km. In a different context, Alvarez Solas et al. (2019, under review) have assessed the sensitivity of the ice-sheet dynamics to the value of the thickness threshold below which the ice is calved through an ensemble exploring a wide value range of this parameter's values, from 10 to 800 meters. Their results showed that the overall effect of this parameter is to modulate the amplitude of the response to the oceanic perturbations but its value did not qualitatively affect their main results. Thus we expect that such a change in the calving law may cause only second-order effects on the retreat.

This point has been reported by adding the sentence in the Discussion section: *"The model behaves sufficiently well in simulating the advance and retreat of the GrIS margins throughout the last 120 kyr (see also Tabone et al (2018)). Part of this performance is related to the two-condition calving law, that is a function of the critical ice thickness $H_f$ below which the ice edge is calved. Thus, depending on its value, this law may be more or less conservative. Here, with an imposed threshold of 200 m, both 79N and ZI floating tongues are lost at the present. It could be that by increasing the value for $H_f$ the model would show slightly more resistance to calve. However, the impact of the calving law is limited to the grid cells at the ice front, while the retreat caused by the submarine melt involves fluctuations in the margin of hundreds of km. Alvarez-Solas et al. (2019, under review) assessed this issue in a different context (the sensitivity of the Eurasian ice sheet to the oceanic forcing during the last glacial period). Their results showed that the overall effect of this parameter is to modulate the amplitude of the response to the oceanic perturbations but its value did not qualitatively affect their main results. Thus, we expect that changes in the calving critical thickness may cause only second-order effects on the retreat."*

The flotation criterion establishes whether the ice at a given grid cell is able to float or must ground. If it floats, the point located immediately inland is assumed to contain the grounding line. Thus, in circumstances of deep bedrock (such as along the 79N and ZI fjords) and high sea level elevation (warmer state), the ice advance could be inhibited if the ice thickness at the front is too low (for dynamical or climatic reasons). This point has also been discussed a few lines below in response to another comment. However, this is more related to the bedrock itself than to the grounding line treatment itself. It would be interesting to perform this study with a more complex grounding-line

treatment to see its effects on the retreat combined with the impact of submarine melt rates. However, again, we expect only second-order effects on the results.

Perhaps a general paragraph on what we know this model is good at in general would therefore be useful.

A paragraph that describes the performance on the model in fitting the present day observations (surface elevation and velocity) has been added in the Discussion to give the reader more detailed information on the model behavior: "*The comparison of our results with observations is a good strategy to assess the model performance and to comprehensively evaluate the robustness of our results. At large spatial scales our simulations fairly represent the present state of the GrIS (Fig. S1). The maximum differences in surface elevations are found in the southwest and in the east due to a mismatch in ice cover. There, the ice sheet ends in many steep and narrow fiords which are not properly represented by the 10km-resolution model. Also, the NEGIS front is located farther inland than as observed. The velocity field shows a pretty good agreement in the interior of the ice sheet, where ice speeds are expected to be lower than 50 m a⁻¹ (Joughin et al., 2018). However, the simulated ice flow of outlet glaciers and ice streams shows more discrepancies. The speed of the inland flow is generally overestimated, whilst the velocities of streams as they extend far inland is underestimated. By zooming into our domain of interest we see that this pattern is also shared by the NEGIS (Fig. S2 and left panel of Fig. S3). The stream geometry is not properly recognized, although the spatial distribution of the velocities is somewhat consistent with observations (faster flow at the margins and reduced speed in the interior, as seen in Fig. S2). However, the tributary fast flows that feed the 79N are not reproduced; the SG is faster than expected and, instead of the long penetrating tongue of ice that characterises the NEGIS, the model simulates a stream catching a wider area.*
*Properly modelling the NEGIS is a well-known problem of ice-sheet models that investigate the evolution of the GrIS at large spatial scales. Most of these models underestimate the stream velocity and do not properly capture its outline (Seddik et al., 2012; Greve and Herzfeld, 2013; Aschwanden et al., 2016; Calov et al., 2018; Golledge et al., 2019). Greve and Otzu (2007) succeed in reproducing a correct magnitude of its speed by increasing the basal sliding under the NEGIS by three orders of magnitude relative to the rest of the ice sheet, but they fail in reproducing its geometry. A good agreement between model and data is found in Price et al. (2011) and Peano et al. (2017), who use a spatially variable basal friction coefficient derived from an iterative inverse method to match the observed velocities.*
*Our imperfect reproduction of the NEGIS stream is probably related to a combination of still low spatial resolution (10 km) and problems in capturing the dynamics at the base of the ice sheet. Our basal friction coefficient β is a function of the effective water pressure at the base of the ice sheet, which is a significant degree of freedom in ice-sheet models. A better representation of basal hydrology and sliding could help to improve the simulation of the ice stream. In parallel, new studies on the origin of the stream (following Roghozina et al., (2016)), its basal characteristics (following e.g. Keisling et al., (2014), Christianson et al. (2014) and Rivermann et al., (2019)) and new data from the EGRIP ice core (following e.g. Vallelonga et al., 2014) will bring new insights in this direction.*"

L14: I was surprised to see that there was no sensitivity testing of the climatic controls on the model experiments. Can you comment somewhere in the text as to whether you think the results would be significantly different if the climate control was altered within a certain range of uncertainty?

The reviewer is right noting that we did not perform sensitivity tests on the climate control. We are aware that this precludes the possibility of assessing the relative role between the atmospheric and oceanic forcings. However, the impact of different climates on the NEGIS margin needs to be evaluated in the future to comprehensively understand the causes of this retreat.

To clarify this point we added this caveat in the results: "*The absence of an equal sensitivity test on the atmospheric forcing, and/or further experiments with another melt scheme, preclude the possibility to assess the relative role between atmospheric and oceanic forcings in the evolution of the NEGIS margin.*"

Also the following paragraph has been added in the Discussion: "*Further experiments accounting for changes in the atmospheric temperatures and precipitation or variations in the external forcing (i.e.*

*insolation) should be carried out for a full understanding of the mechanisms involved in this retreat, here explained by considering the sole impact of the ocean. Particularly, a sensitivity study on climatic variations performed with a prescribed ocean could help constraining the effect of the atmosphere in this phenomenon to eventually evaluate the relative role between the forcings in driving the NEGIS margin."*

Page 4:

L18: Can you justify why basal melt rate is not parameterised in a depth-dependent manner? Or in other words, why is a 10% Bm on all floating grid cells an appropriate decision to make?

To make the experiment as simple as possible and within the scope of a sensitivity study, we impose a spatially homogeneous temperature anomaly at the base of the ice shelf that is the 10% of that at the grounding line. This of course is a simplification of reality since submarine metl rates vary at regional and local spatial scales. However, this assumption is in line with present-day estimates of submarine melt rates at the margins of the GrIS (e.g. Münchow et al., 2014; Rignot and Jacobs, 2002; Wilson et al., 2017).

A similar approach has been also used in previous work (Alvarez-Solas et al., 2019, Blasco et al., 2019, Tabone et al., 2018) investigating the past role of the ocean on the GrIS evolution during the last two glacial cycles. A more realistic approach considering depth-dependent submarine melt rates also that spatially vary across the Greenland coasts is beyond the scope of our simple sensitivity study. However, these features should be taken into account in the scope of future work.

L25 or thereabouts: Do you make any assessment of whether the pre MIS3 state is realistic? I.e. how good is the spin-up, can it be assessed, and how does it fit to the geological/field data from the region. In addition, can you confirm you are you allowing the grounding line to evolve through time? And does the ice shelf characteristics evolve through time during spin-up?
L30: I think a map of the fit between the pre MIS-3 state and the geological evidence would be an important figure to show. This will allow better discussion of whether the system is appropriately setup for the retreat experiments. I.e. if the extent or thickness is not correct, then how can we trust the degree or rates of retreat?

The two issues above are addressed together. The spin-up covers the penultimate glacial cycle (between 250 kyr and 120 kyr ago); it includes a variable topography, variable climatology, the grounding line migration is enabled, and the submarine melting is defined through the same equation as the variable climatology (i.e. follows the orbital-driven temperature evolution, extrapolated from Kindler et al., 2014 up to 300 kyr ago). There is actually no difference in the experimental design between the spin-up and the rest of the simulation, but that interval of time is not analysed but only used to allow the model to stabilise after the initialisation. Results for the GrIS at the LIG are well within the range of other reconstructions, as results are very similar to those of Tabone et al. (2018).

However, it is hard to discuss in detail these spin-up results since, besides the model reconstructions available for the LIG, there is a huge uncertainty about the GrIS configuration before the LIG and between the LIG and the LGM. Specifically, as far as we know there is no such geological evidence for the NEGIS margin configuration before the time period showed in Larsen et al. (2018) (ca. 41 ka BP). Although temperature reconstructions, accumulation changes, insolation changes, etc. suggest that the GrIS between the LIG and MIS-3 was in a glacial state (e.g. Dahl-Jensen et al. (1998), Andersen et al. (2004), NEEM community members (2013)) and many Greenland marine-terminating glaciers advanced over the shelf reaching at least the inner part of it (e.g. Funder et al., 2011), we do not know if the NEGIS margin was actually expanded toward the continental shelf during that time, and if yes, to what extent.

Since we don't know the state of the NEGIS before MIS-3 due to the lack of data, we cannot actually corroborate that there had been a retreat from a pre-MIS-3 to a MIS-3 state. On the contrary, we want to emphasize that we can simulate a margin that *was located* farther inland during part of MIS-3 and advanced toward the shelf break during the LGM, as suggested by reconstructions. More generally, our results are in agreement with those from Larsen et al. (2018), suggesting that the

NEGIS margin strongly fluctuated during the last glacial period, but we provide a different explanation which we think is more realistic.

However a similar assessment can be made by comparing our results with the geological records for the LGM. Since our results are well in agreement with the available reconstructions (between 41 kyr BP and the present, as seen in new Fig 3) and the spin up for the first glacial cycle reproduces ranges of advance/retreat in the range of the uncertainties (not shown), we are pretty confident in the reliability of our results. However, this is discussed in more detail below, in a response to a related comment.

L32: delete 'already'.

Done.

Page 5:

L1: What is meant by 'substantially steady' – describe the margin stability pattern in clearer detail.

This sentence has been changed to: "*During MIS-3, the ice-margin position gradually advances towards its maximum glacial extent, which is reached at about 20 kyr BP (LGM), when the ice sheet becomes grounded at about 30 km far from the shelf break, reducing the area of the floating ice shelf in the region (Fig 4 a-e)."*

Note that old Fig. 1 has been changed with new Fig. 3, and old Fig.3 with new Fig. 4.

L21: 'stationing' isn't a good word to use. Do you mean 'stopping', or 'retreating to'?

Yes, we agree with the reviewer that using the word "stopping" is more appropriate. Changed accordingly.

L27: you mention that there is no melt imposed at the LGM. You could discuss somewhere later about whether you think this is a realistic condition.

This point has been addressed adding this sentence in the Methods: "*Imposing a submarine melting rate equal to zero at the LGM is probably a simplification of reality, leading to the absence of refreezing below the ice shelves. These processes may vary strongly at local scales, as we know from present observations in Antarctica (e.g. Rignot et al., 2013) and Greenland (e.g. Wilson et al., 2017). However due to the lack of data for the basal melt along the NEGIS margins for the last glacial and the coarse resolution of our model (10 km), this assumption may be considered as a fair compromise for the scope of our sensitivity test."*

Page 6:

L1 'saturates for high values' – can the high values be stated quantitatively?

This sentence has been changed to: *"This feature then saturates for Bref higher than 3 m a$^{-1}$, as a further retreat inland is constrained by the bathymetry."*

L14-19: As mentioned before – I would like to see a better exploration, and a figure, showing the fit of the LGM expansion and the field data. I think the weakness of this paper lies in both a lack of description of this, but also the fact that the fit is not as good as it could be. We really need to see how good the fit is so we can better judge the results. In addition, were there no modifications, for example to the climate, or the the Bm during the advance phase, that would help enable a better fit to the data? Some more sensitivity tests on this would have been good to see.

In the new version of the manuscript we compare our simulated maximum glacial extent to previous reconstructions based on geological records (Evans et al., 2009; Arndt et al., 2015, 2017; Winkelmann et al., 2010, Funder et al., 2011) and the results are fairly good, specifically if compared

to previous model reconstructions (new Fig. 7). However, there are still some discrepancies between model and data, specifically we cannot advance the grounding line up to the continental shelf break. Here we discuss what we think might be responsible for the limited extent and the consequent constrained ice volume at the LGM.

We changed the paragraph of Pag. 6 lines 14-19 (old MS) to:

*"Second, our simulated grounding-line advance during the LGM is smaller that the maximum extension suggested by reconstructions from geological records (Fig. 7). This bias furthermore increases with increasing oceanic forcing. Even in the unperturbed experiment, which allows the largest ice-sheet expansion due to the absence of melting at the marine margins, the grounding-line does not reach the continental shelf break either. Nevertheless, our simulated extent is still one of the best reconstructions of the northeast Greenland at the LGM obtained with an ice sheet model (Tabone et al., 2018, Simpson et al., 2009, Lecavalier et al., 2014, Bradley et al., 2018). This discrepancy in the LGM extents is reflected in the transient GrIS sea level contribution from the LGM to the present (Fig. S4), that is underestimated as compared to other recent modelling work (Lecavalier et al., 2014, Tabone et al., 2018). Nevertheless, our estimation is not far from others (Simpson et al., 2009, Huybrechts 2002) and well within the range proposed by Buizert et al. (2018). Note that although the LGM extent simulated by Lecavalier et al (2014) is smaller than ours in the northeast, their ice volume contribution at the glacial maximum is about 1 m SLE higher. This could be partly due to their larger grounding-line advance in the northwest, but it might be also related to a more active dynamics in our simulations. The volume discrepancy between our two studies performed using the same ice-sheet-shelf model are likely due to differences in the dynamics. The main reason seems to be related to the fact that SIA and SSA velocities are here simply summed up instead of mixed through a weighting function as in Tabone et al., 2018. This increases the velocities in the transition zones, promoting discharge of ice from the interior and consequently limiting the ice volume accretion. Second, Tabone et al. (2018) accounted for refreezing processes at the base of the ice shelves, which allowed the grounding line to advance easily, leading to a glacial state in which almost all the GrIS margins were able to reach the shelf break. It is clear that this larger extent could account for a substantial part of the ice volume discrepancy. Another possible reason could be that here we increased the basal drag at the base of grounded temperate ice (by increasing its coefficient $c_f$ of Eq. 2). More friction at the base may foster the production of water at the ice-bed interface through heat release, making the bed more slippery and the ice flow to accelerate. However, we expect that this process is responsible for only a small fraction of the ice volume discrepancy, since it is counteracted by the increase in basal friction itself. Increasing the total ice volume during the glacial (and its extent) would probably require a substantial tuning effort, that is beyond the scope of this study. Our goal is not to provide a perfect match with the LGM but to illustrate a plausible mechanism behind the retreated ice margin at MIS-3 and its subsequent advance."*

L17: 'insufficient basal drag' . Please tell us why the imperfect drag imposition would alter the result in this way?

Please, see the paragraph above.

Page 7:

L1: can you describe the pattern of saturation in more quantitative detail?

The paragraph has been changed to: *"The grounding-line retreat at the PD is proportional to the magnitude of the submarine melt rate imposed at the NEGIS ice margins during the mid-late Holocene, which is related to the value of Bref used. However, this correspondence is very weak and the retreat quickly saturates at about 70 km away from the PD position along the glacier flowing direction for κ higher than 3 m a-1 K-1, where it is stopped by the presence of a bedrock above the sea level (Schaffer et al., 2016; Morlighem et al., 2017). Although this retreat is supported by proxies for the mid-Holocene (Bennike and Weidick, 2001; Larsen et al., 2018), its persistence until the present day is unrealistic. "*

L9: 'it is unlikely that this could have happened for a long period of time and in such a persistent way.' Can you justify why this is the case? Explain in more detail – link to any knowledge in ocean circulation change etc.

*This sentence has been added in the discussion:* "Peaks of up to 50 m a−1 occur at the NEGIS margin, however it is unlikely that this could have happened for a long period of time and in such a persistent way. Several records inferred from sediment cores in the Arctic Ocean and in Fram Strait indicate that temperatures of surface and subsurface waters strongly fluctuated during the Holocene due to the variability of the oceanic currents. The inflow of Atlantic warming waters in the early Holocene determined warmer oceanic conditions recorded at the surface (Sztybor and Rasmussen 2017) and at the subsurface (Werner et al., 2016, 2013), where temperatures increased by 3-4 °C since the beginning of the Holocene. After 9-8 kyr however, these records report a drop in temperatures, gradually (Falardeau et al., 2018.; Werner et al., 2016) or interrupted by some peaks of warming (Consolaro et al. 2018). These different oceanic conditions between the early and mid-late Holocene suggest that such a durable high melting rate during the whole Holocene is likely overestimated. "

Page 8:

L4: 'helps to constrain' – so you mean 'helps to limit'? Constrain could be interpreted in a number of ways – e.g. to limit or to provide evidence to help understand.

Changed to "helps to limit".

L7: It would be useful to know whether the 'prolonged presence of submarine melt' is something that is a realistic prospect based on any other evidence.

*This concern has been addressed in the discussion by adding this paragraph:* "Paleoceanographic records inferred from marine sediments in the Arctic Sea and Fram Strait that provide information on the oceanic state during MIS-3 at high temporal resolution are scarce. However, they all suggest rapid temperature fluctuations as a result of large changes in water masses at different depths. Warmer SST may last for 3-4 kyr before cooling (Muller et al., 2014). Generally, strong variations in the oceanic conditions are found between glacial-interglacial, but also between larger stadial-interstadial transitions (Poirer et al., 2012). A sediment record between the Nordic seas and the Arctic Ocean suggest that high SST and low intermediate water temperatures are typical of interstadials, while the opposite is found during stadials due to intrusion of warmer Atlantic subsurface water (Rasmussen et al., 2014). This strong oceanic temperature variability during the last 50 kyr is also documented by another record based on a stack of sediment records of the Arctic Ocean and the Fram Strait, suggesting the occurrence of several peaks of warmings during MIS-3 reaching temperatures 1-3 °C higher than those recorded for the Holocene (Cronin et al., 2012). However, a qualitative analysis of this temperature record at long (orbital) timescales indicate that its evolution agrees well with that of the melting rate signal used in this work: high melting during MIS-3, prolonged cooling during the LGM and high melting again during the Holocene. Thus, even though we remove some degree of realism by not considering the millennial-scale variability in the ocean, our experimental design could fairly represent the evolution of northern Greenland oceanic conditions at long timescales."

Figures:

Fig1: I would separate the inset map to a separate new figure 1 which should be a location map showing the NEGIS area in much more detail, including the key outlets, the location of the profiles, the offshore bathymetry, the onshore ice cover etc. In addition, either on that map or on an additional new figure, all the evidence for past ice extent should be shown so that we can then use it to judge how well the model fits with
the geological dataset.

As already said at the beginning of this document, we added in the MS a map addressing all these points (new Fig. 2) in substitution of the inset map of old Fig. 1.

In addition, in this figure the growth and retreat steps are rapid. Can you say much about whether these are purely a function of the forcing provided, or whether the bed topography or fjord width is having any particular control on the 'stepped' nature of advance or retreat?

The "stepped" nature of advance and retreat, as the reviewer calls it, is primarily due to the oceanic forcing applied. This is pretty well visible comparing the submarine melting rate signal with the curve of old Fig. 1 (new Fig. 3). The large advance and retreat well follow the decreasing/increasing in melt. However, others factor may also play a role. First, the area connecting the ZI and 79N to the inland topography shows a bedrock that is 100-200 m deep (Schaffer et al., 2017; Morlighem et al., 2017). In periods of relatively high sea level, such as the first thousand years after the last interglacial, this deep bathymetry may be crucial in driving the grounding line evolution (through the flotation criterion), since it hampers the ice to ground and so the ice sheet to advance. Another reason for this "stepped" nature is the fact that in the old version of the MS we calculated the grounding-line position in time over one transect only, thus per each grid point "conquered" by ice the grounding line would advance of 10 km in a row.

To smooth the results, we now calculate the grounding-line distance on 48 transects intersecting the ZI and the continental shelf break. The final result is the average of the grounding-line position between all of them (new Fig. 3). This improvement in the experimental setup allows for a better characterisation of the results, since 1) the big uncertainty in the LGM grounding-line position of Larsen et al. (2018) (+- 50 km) accounts for the broad range of distances between the shelf break and the PD margin of the outlet glaciers and 2) the retreated position during MIS-3 is documented only for ZI (Larsen et al., 2018) and SG (Weidick et al., 1996).
This helps to smooth the grounding-line distance, however the "stepped" nature related to the other two sources explained above is of course still visible.

To make these aspects clear this sentence has been added to the results: *"We calculate the grounding-line distance from the PD position on 48 transects intersecting the ZI and the continental shelf break (Fig. 2). Then we average the results to create one transient evolution for the grounding line for each oceanic forcing."*

Also: *"Grounding-line advance and retreat is often very rapid, especially during the first advance after the LIG or during the MIS-3. This is primarily due to the oceanic forcing applied, since the large advance and retreat well follow submarine melt rate evolution. Part of this stepped nature may be due to the bathymetry too. The area connecting the ZI and 79N to the inland topography shows a bedrock 100-200 m deep (Morlighem et al., 2017). In periods of relatively high sea level, such as the first kyr after the last interglacial, this deep bathymetry may be crucial in driving the grounding line evolution (in our model through the flotation criterion), since it hampers the ice to ground and so the ice sheet to advance. This is in line with recent work suggesting that deep bathymetry combined with warmer waters entering the fjord may have important consequences in the destabilisation of the 79N (Schaffer et al., 2017)."*

Given that you mention that there was a retreat to a position inland of the present day grounding line position, can you also show a horizontal dotted line to represent the knowledge of where this inland retreat reached?

We do not understand to what position the reviewer is referring. The position of the grounding line during MIS-3 as reconstructed by Larsen et al., (2018) is represented by the dashed black line between 41-26 kyr BP in new Fig. 3 (old Fig. 1). The position reached by our simulated grounding line during the Holocene is shown by the curves representing different oceanic forcings.

Fig3: These figures are a little hard to follow because of their size. Can they be made bigger? The arrows pointing to the PG grounding line positions aren't terribly useful – surely a line on the map would be more appropriate. Finally, you mention these are snapshots at different times along MIS3 and the LGM. Please state which times these actually represent.

New Fig. 4 (old Fig. 3) is now bigger; the arrows indicating the PD grounding line have been substituted with a thin black curve and the snapshots have been named by the time they represent. Also, the reconstructed LGM extent is added to each snapshot for comparison.

REFERENCES:

Alvarez-Solas,et al., 2018: Oceanic forcing of the Eurasian Ice Sheet on millennial time scales during the Last Glacial Period, Clim. Past Discuss., https://doi.org/10.5194/cp-2018-89, in review, 2018.

Andersen et al. 2004: "High-resolution record of Northern Hemisphere climate extending into the last interglacial period." Nature 431.7005 (2004): 147.

Arndt et al., 2015. A new bathymetry of the Northeast Greenland continental shelf: Constraints on glacial and other processes. *Geochemistry, Geophysics, Geosystems,* 16, 10 (2015): 3733-3753.

Arndt et al., 2017. The last glaciation and deglaciation of the Northeast Greenland continental shelf revealed by hydro-acoustic data. *Quaternary Science Reviews,* 160 (2017): 45-56.

Aschwanden et al., 2016. Complex Greenland outlet glacier flow captured. *Nature communications* 7 (2016): 10524.

Bennike and Weidick, 2001. Late Quaternary history around Nioghalvfjerdsfjorden and Jøkelbugten, North-East Greenland. *Boreas* 30.3 (2001): 205-227.

Blasco et al., 2019. The Antarctic Ice Sheet response to glacial millennial-scale variability. Climate of the Past 15.1 (2019): 121-133.

Bradley, Sarah L., et al. "Simulation of the Greenland Ice Sheet over two glacial-interglacial cycles: investigating a sub-ice-shelf melt parameterization and relative sea level forcing in an ice-sheet-ice-shelf model." Climate of the Past 14.5 (2018): 619-635.

Buizert et al., 2018 "Greenland-Wide Seasonal Temperatures During the Last Deglaciation." Geophysical Research Letters 45.4 (2018): 1905-1914.

Calov et al., 2018. Simulation of the future sea level contribution of Greenland with a new glacial system model. The Cryosphere 12.10 (2018): 3097-3121.

Choi et al., 2017. Modeling the response of Nioghalvfjerdsfjorden and Zachariae Isstrøm Glaciers, Greenland, to ocean forcing over the next century. Geophysical Research Letters 44.21 (2017): 11-071.

Christianson et al. (2014). Dilatant till facilitates ice-stream flow in northeast Greenland. Earth and Planetary Science Letters 401 (2014): 57-69.

Colleoni et al., 2014. Modeling Northern Hemisphere ice-sheet distribution during MIS 5 and MIS 7 glacial inceptions. Climate of the Past 10.1 (2014): 269-291.

Consolaro 2018. Palaeoceanographic and environmental changes in the eastern Fram Strait during the last 14,000 years based on benthic and planktonic foraminifera.

Cronin 2012. Deep Arctic Ocean warming during the last glacial cycle. estimate intermediate water temperatures over the past 50,000 years from the Mg/Ca from Arctic sediment cores.

Dahl-Jensen et al., 1998. Past temperatures directly from the Greenland Ice Sheet. Science 282, 268-271.

Evans et al., 2009. Marine geophysical evidence for former expansion and flow of the Greenland Ice Sheet across the north-east Greenland continental shelf. Journal of Quaternary Science: Published for the Quaternary Research Association, 24, 3 (2009): 279-293.

Falardeau 2018. Paleoceanography of northeastern Fram Strait since the last glacial maximum: Palynological evidence of large amplitude changes.

Funder, Svend, et al. "The Greenland Ice Sheet during the past 300,000 years: A review." Developments in Quaternary Sciences. Vol. 15. Elsevier, 2011. 699-713.

Golledge et al., 2019. Global environmental consequences of twenty-first-century ice-sheet melt. Nature 566.7742 (2019): 65.

Greve and Herzfeld, 2013. Resolution of ice streams and outlet glaciers in large-scale simulations of the Greenland ice sheet. Annals of Glaciology 54.63 (2013): 209-220.

Greve and Otzu, 2007. The effect of the north-east ice stream on the Greenland ice sheet in changing climates. The Cryosphere Discussions 1.1 (2007): 41-76.

Huybrechts, Philippe. "Sea-level changes at the LGM from ice-dynamic reconstructions of the Greenland and Antarctic ice sheets during the glacial cycles." Quaternary Science Reviews 21.1-3 (2002): 203-231.

Joughin et al., 2018. Greenland Ice Mapping Project: ice flow velocity variation at sub-monthly to decadal timescales.

Keisling et al, 2014. "Basal conditions and ice dynamics inferred from radar-derived internal stratigraphy of the northeast Greenland ice stream." Annals of Glaciology 55.67 (2014): 127-137.

Larsen et al., 2018. Instability of the Northeast Greenland Ice Stream over the last 45,000 years. Nature communications 9.1 (2018): 1872.

Lecavalier et al., 2014. A model of Greenland ice sheet deglaciation constrained by observations of relative sea level and ice extent. Quaternary Science Reviews 102 (2014): 54-84.

Mayer et al., 2018. Large ice loss variability at Nioghalvfjerdsfjorden Glacier, Northeast-Greenland. Nature communications 9.1 (2018): 2768.

Morlighem, Mathieu, et al. "BedMachine v3: Complete bed topography and ocean bathymetry mapping of Greenland from multibeam echo sounding combined with mass conservation." Geophysical Research Letters 44.21 (2017): 11-051.

Mouginot et al., 2015. Fast retreat of Zachariæ Isstrøm, northeast Greenland. Science 350.6266 (2015): 1357-1361.

Müller et al., 2014. High-resolution record of late glacial and deglacial sea ice changes in Fram Strait corroborates ice–ocean interactions during abrupt climate shifts.

Münchow, A., Padman, L., and Fricker, H. A.: Interannual changes of the floating ice shelf of Petermann Gletscher, North Greenland, from 2000 to 2012, Journal of Glaciology, 60, 489–499, 2014.

NEEM community members, 2013. Eemian interglacial reconstructed from a Greenland folded ice core. Nature 493.7433 (2013): 489.

Peano et al., 2017: "Ice flux evolution in fast flowing areas of the Greenland ice sheet over the 20th and 21st centuries." Journal of Glaciology 63.239 (2017): 499-513.

Peyaud, V., C. Ritz, and G. Krinner. "Modelling the Early Weichselian Eurasian Ice Sheets: role of ice shelves and influence of ice-dammed lakes." Climate of the Past Discussions 3.1 (2007): 221-247.

Poirier 2012.Central Arctic paleoceanography for the last 50 kyr based on ostracode faunal assemblages.

Price, Stephen F., et al. "Committed sea-level rise for the next century from Greenland ice sheet dynamics during the past decade." Proceedings of the National Academy of Sciences 108.22 (2011): 8978-8983.

Rasmussen et al., 2014. , Water mass exchange between the Nordic seas and the Arctic Ocean on millennial timescales during MIS 4–MIS 2.

Rathmann et al., 2017. Highly temporally resolved response to seasonal surface melt of the Zachariae and 79N outlet glaciers in northeast Greenland. Geophysical Research Letters 44.19 (2017): 9805-9814.

Rignot, E. and Jacobs, S. S.: Rapid bottom melting widespread near Antarctic Ice Sheet grounding lines, Science, 296, 2020–2023, 2002.

Rignot, E., et al 2013:. "Ice-shelf melting around Antarctica." Science 341.6143 (2013): 266-270.

Riverman et al., 2019. Riverman, K. L., et al. "Enhanced Firn Densification in High-Accumulation Shear Margins of the NE Greenland Ice Stream." Journal of Geophysical Research: Earth Surface 124.2 (2019): 365-382.

Roghozina et al., 2016. Melting at the base of the Greenland ice sheet explained by Iceland hotspot history. Nature Geoscience 9.5 (2016): 366.

Schaffer et al., 2016. A global, high-resolution data set of ice sheet topography, cavity geometry, and ocean bathymetry. (2016): 543-557.

Seddik et al., 2012. Simulations of the Greenland ice sheet 100 years into the future with the full Stokes model Elmer/Ice. Journal of Glaciology 58.209 (2012): 427-440.

Simpson, Matthew JR, et al. "Calibrating a glaciological model of the Greenland ice sheet from the Last Glacial Maximum to present-day using field observations of relative sea level and ice extent." Quaternary Science Reviews 28.17-18 (2009): 1631-1657.

Sztybor and Rasmussen 2017. Late glacial and deglacial palaeoceanographic changes at Vestnesa Ridge, Fram Strait: Methane seep versus non-seep environments.

Tabone et al., 2018. The sensitivity of the Greenland Ice Sheet to glacial-interglacial oceanic forcing, Clim. Past, 14 (2018): 455-472.

Vallelonga et al., 2014. Initial results from geophysical surveys and shallow coring of the Northeast Greenland Ice Stream (NEGIS). The Cryosphere 8.4 (2014): 1275-1287.

Weidick et al., 1996. Neoglacial glacier changes around Storstrommen, North-East Greenland. Polarforschung 64.3 (1996): 95-108.

Werner et al., 2013. Atlantic water advection to the eastern Fram Strait during the last 9 ka: multiproxy evidence for a two-phase Holocene. Paleoceanography 28, 283-295.

Werner, K. et al. Holocene sea subsurface and surface water masses in the Fram Strait - comparisons of temperature and sea-ice reconstructions. Quat. Sci. Rev. 147, 194–209 (2016).

Wilson, et al., 2017: "Satellite-derived submarine melt rates and mass balance (2011–2015) for Greenland's largest remaining ice tongues." (2017).

Winkelmann et al., 2010. Submarine end moraines on the continental shelf off NE Greenland–Implications for Lateglacial dynamics. *Quaternary Science Reviews,* 29, 9-10 (2010): 1069-1077.

FIGURES

[Figure]

New Fig. 2. Map of the NEGIS sector showing the location of its three outlet glaciers (79N, ZI and SG), the observed present grounding-line position (solid black line), the observed present surface

velocities (from Joughin et al., 2018), the offshore bathymetry and the onshore ice cover (both from Schaffer et al., 2016) and the maximum (dotted black line) and minimum (dashed black line) grounding-line positions reconstructed for the LGM (Funder et al., 2011). The 48 transects used to calculate the evolution of the grounding-line position are shown in purple.

[Figure]

New Fig. 7. Simulated GrIS extent at the LGM for different oceanic forcings compared to other glacial reconstructions. Colored lines follow the color scale of new Fig. 3 in the MS (old Fig. 1). The solid

black line refers to the maximum glacial extent simulated by Lecavalier et al. (2014), calibrated to match the minimum LGM configuration (Funder et al., 2011) in the northeast. The dashed black line represents the expected maximum glacial extent at the northeast sector as inferred from various geological data (Evans et al., 2009; Arndt et al., 2015, 2017; Winkelmann et al., 2010).

[Figure]

New Fig. S1 in the MS. Simulated minus observed GrIS surface elevation (left panel) and GrIS ice velocity (right panel) for the PD. Green and black lines on the left represent simulated and observed GrIS extents, respectively. Surface elevation data are taken from Schaffer et al. (2016); ice velocity observations from Joughin et al. (2018). Both maps are produced for the $\kappa = 8$ m a$^{-1}$ K$^{-1}$ experiment. However, the choice of another oceanic sensitivity $\kappa$ would have little effect on the simulated-observed discrepancy.

[Figure]

New Fig. S2 in the MS. Present-day simulated (left panel) and observed (right panel) velocities for the NEGIS sector. Observed data are taken from Joughin et al. (2018).

[Figure]

New Fig. S3 in the MS. Simulated-observed present-day velocities for the NEGIS sector (left panel) and its scatterplot (right panel). Blue line refers to the perfect match between model and data.

[Figure]

New Fig. S4 in the MS. Evolution of the GrIS sea-level contribution for the last 60 kyr. Colored curves refer to the color scale of new Fig. 3 in the MS (old Fig. 1). The black curve refers to the GrIS sea level contribution for the last deglaciation modelled by Lecavalier et al. (2014).

---

## Author Comment (AC2) · 9 Apr 2019

The objective of the study is to test whether submarine melt (ocean warming) could be the primary cause of the ice margin retreat of NEGIS during MIS3 and MIS1 that was recently documented by Larsen et al 2018 using radiocarbon dating of reworked shells in historical (LIA) moraines. It uses the GRISLI-UCM 3D ice-sheet-shelf model to simulate the influence of submarine melt using a variable amount of melt rates. I am not an expert in ice sheet modelling and cannot evaluate if the model set-up is state-of-the-art, but the description of the model set-up is easy to follow and understandable. It also seems to be realistic melt-rates that have been used to force the model. The manuscript is generally well-written, and the model-data comparison provides new and interesting knowledge about the potential effect of ocean warming and submarine melt on the evolution of NEGIS. However, there are a few places where minor revision is warranted. These are listed below.

We are grateful to the reviewer for their positive evaluation of our work. Answers to their specific comments are reported below.

Title: I am not aware of the TC politics on using abbreviations in the title, but I would avoid using them. The title could be changed to: Submarine melt as a potential trigger of ice margin retreat of the Northeast Greenland Ice Stream during Marine Isotope Stage 3.

Title changed to: "*Submarine melt as a potential trigger of ice margin retreat of the Northeast Greenland Ice Stream during Marine Isotope Stage 3.*"

L1: Remove "area"

Done.

L5: Why is this a conundrum? – this should be explained in more detail.

We agree that this paragraph needs clarification. We changed it to: "*Alongside, a recent study suggests that the NEGIS grounding line was 20-40 km behind its present-day location for 15 ka during Marine Isotope Stage (MIS) 3. This is in contrast with Greenland temperature records indicating cold atmospheric conditions at that time, expected to favor ice-sheet expansion. To explain this anomalous retreat a combination of atmospheric and external forcings has been invoked. However, the ocean was not brought into play. Here we investigate the sensitivity of the NEGIS to the oceanic forcing during the Last Glacial Period (LGP) using a three-dimensional hybrid ice-sheet-shelf model. We find that a sufficiently high oceanic forcing could account for a NEGIS ice-margin retreat of several tens of km, potentially explaining the recently proposed NEGIS grounding-line retreat during MIS-3.*"

L9: MIS-3 = Marine Isotope Stage 3

Changed accordingly.

L11-: change to.. . . ..even retreating 70 km behind its present-day position from 7.8-1.2 ka during most of the mid- and late Holocene and 20-40 km from 41-26 ka during Marine Isotope Stage 3 (MIS-3, c. 60-25 ka).

Combining this and other reviewers' suggestions this sentence has been changed to: "*Around 41-26 ka BP during Marine Isotope Stage 3 (MIS-3, c. 60-25 ka) the NEGIS front was ca. 20-40 km farther*

*inland than today, then advanced by more than 250 km toward the shelf break at the Last Glacial Maximum (LGM) and retreated again during the last deglaciation, at ca. 70 km behind its present-day position, where stopped most of the mid-and late Holocene (7.8-1.2 ka BP).* "

L12: Stage NOT state

Changed accordingly. It was a typo.

L20: change to (LIG, c. 128-116 ka)

Changed accordingly.

L15: change to (c. 116 ka)

Changed accordingly.

L20 and L24: I guess it should be the last 45 ka?

We agree with the reviewer, L20 was "45 kyr". Changed accordingly. In L24, however, we were actually meaning "35 kyr" since we were referring to the evolution of the submarine melting forcing, which by construction is cut off at zero at 35 kyr.

L34: change to mid- and late Holocene

Changed accordingly.

Figure 1: I would make the inset map bigger and outline NEGIS – maybe as a panel next to the diagram. It would also be useful if the LIG, MIS3, LGM, Holocene time periods are as shown as vertical bars.

These suggestions have been taken into account when producing the new Fig. 3 (old Fig. 1). The inset map has been substituted by a new stand-alone figure zoomed on the analysed sector showing the location of the outlet glaciers, the observed present grounding-line position, the LGM reconstructed grounding-line positions (max and min), PD observed surface velocities, offshore bathymetry and the onshore ice cover (new Fig. 2).

Figure 3: I would suggest making the figure bigger as it is difficult to see the details in the maps. Maybe an outline of NEGIS could be placed on top of the velocity fields? It would also be valuable for the discussion if the LGM reconstruction of Funder et al and the minimum reconstructions (MIS-3 and MIS1) of Larsen et al could be shown on the maps.

New Fig. 4 (old Fig. 3) is now bigger; the arrows indicating the PD grounding line have been replaced by a thin black curve and the snapshots have been named by the time they represent. Also, the reconstructed LGM extent from Funder et al., 2011 (based on Evans et al., 2009; Arndt et al., 2015, 201; Winkelmann et al., 2010) is added to each snapshot for comparison. Since the minimum extent during MIS-3 provided by Larsen et al. (2018) is inferred from only a couple of locations close to ZI and SG, tracing a 2D prospect of the grounding-line position during MIS-3 would be hard from the limited information on the retreat coming from that work, we decided to not include the MIS-3 minimum reconstruction on the map.

**REFERENCES:**

Arndt et al., 2015. A new bathymetry of the Northeast Greenland continental shelf: Constraints on glacial and other processes. *Geochemistry, Geophysics, Geosystems,* 16, 10 (2015): 3733-3753.

Arndt et al., 2017. The last glaciation and deglaciation of the Northeast Greenland continental shelf revealed by hydro-acoustic data. *Quaternary Science Reviews,* 160 (2017): 45-56.

Evans et al., 2009. Marine geophysical evidence for former expansion and flow of the Greenland Ice Sheet across the north-east Greenland continental shelf. *Journal of Quaternary Science: Published for the Quaternary Research Association,* 24, 3 (2009): 279-293.

Funder, Svend, et al. "The Greenland Ice Sheet during the past 300,000 years: A review." *Developments in Quaternary Sciences*. Vol. 15. Elsevier, 2011. 699-713.

Larsen et al., 2018. Instability of the Northeast Greenland Ice Stream over the last 45,000 years. *Nature communications* 9.1 (2018): 1872.

Winkelmann et al., 2010. Submarine end moraines on the continental shelf off NE Greenland–Implications for Late glacial dynamics. *Quaternary Science Reviews,* 29, 9-10 (2010): 1069-1077.

[Figure]

New Fig. 2. Map of the NEGIS sector showing the location of its three outlet glaciers (79N, ZI and SG), the observed present grounding-line position (solid black line), the observed present surface velocities (from Joughin et al., 2018), the offshore bathymetry and the onshore ice cover (both from Schaffer et al., 2016) and the maximum (dotted black line) and minimum (dashed black line) grounding-line positions reconstructed for the LGM (Funder et al., 2011). The 48 transects used to calculate the evolution of the grounding-line position are shown in purple.

[Figure]

New Fig. 2 (old Fig. 1). Evolution of the NEGIS grounding line relative to its observed present-day position for the set of experiments. The grounding-line distance has been calculated along 48 transects which follow approximately the flow direction of NEGIS ZI glacier towards the shelf break (Fig. 2). Dashed black line shows the reconstruction by Larsen et al. (2018). Shaded regions represent the time periods relative to the LIG, MIS-3 and the Holocene. The three dotted curves show the PD NEGIS grounding-line position (0 km), the maximum (300 km +- 50 km) expected advance of the northeastern part of the ice sheet at the LGM according to Funder et al. (2011) and Larsen et al., (2018).

[Figure]

New Fig. 4 (old Fig. 3). Snapshots of U (m a−1) in total absence of submarine melting (a-e) and in presence of active orbital-driven oceanic forcing (κ = 8 m a−1 K−1 , Bref = 8 m a−1 ) (f-j) at different times along MIS-3 and the LGM. The black line represents the position of the simulated grounding line. Grey thin solid line represents the observed PD grounding-line position (Schaffer et al., 2016). Maximum (dotted black line) and minimum (dashed black line) grounding-line positions reconstructed for the LGM (Funder et al., 2011) are also shown.

---

## Author Comment (AC3) · 9 Apr 2019

The study of Tabone et al. focuses on the Northeast Greenland Ice Stream (NEGIS) and its response to changes in climate, and in particular submarine melt, during the last glacial period. By applying climate forcing mimicking conditions during the last glacial period, an ice sheet/ice shelf model is used to study the transient evolution of the Greenland ice sheet over the past 120ka years. The evolution of the NEGIS are discussed in light of existing reconstructions of its history.

The study is original in assessing the long term response of the NEGIS to changes in climate, and goes beyond state of the art by comparing the dynamical evolution of the ice stream to proxy records. The paper is well written and the figures are clear. However, there are a several concerns which should be considered before publication in the cryosphere.

We are glad the reviewer valued our work and we thank them for their constructive comments and suggestions that certainly helped to make the manuscript (MS) clearer and more exhaustive. Answers to general and specific comments are reported below.

GENERAL COMMENTS:

The results of the study are clearly novel and of great potential in our understanding of the long term evolution of the NEGIS. However, there is a lack of detail in the description of the model results and the full potential of the study is untapped.

Given that the model simulates the entire Greenland Ice sheet these results should be included and discussed. In particular, how well does the model reproduce the present data ice sheet configuration as well as the ice stream. Similarly, how do the model results compare to published simulations and reconstructions of the LGM configuration of Greenland. This should also include an assessment of the transient evolution of the equivalent sea level contribution from Greenland.

For the NEGIS it is not clear how well the ice stream itself is reproduced by the model. To what extent does the model capture the observed geometry and velocities of the ice stream? And in particular, an assessment of the time evolution of the ice stream should be included. In what periods was the ice stream active, and did it change its position through time? If possible the model simulations should be compared with reconstructions from marine sediment archives. To make these comparisons relevant, as more data on the evolution of NEGIS become available, a time series showing the simulated ice flux at the margin of NEGIS should be included.

We agree with the reviewer that we might have been too concise in describing the overall behavior of our ice-sheet-shelf model. The MS was missing a detailed description of how the model is able to reproduce the past and present evolution of the entire Greenland Ice Sheet (GrIS). In response to this, we implemented the MS with a new series of figures that should satisfactorily answer the points raised by the reviewer. Some of these figures have been included as Supplementary Material to limit the MS length.

Specifically, we introduced:
- a comparison between simulated-observed present-day (PD) surface elevation (data from Schaffer et al., 2016) and ice velocities (data from Joughin et al., 2018) for the entire Greenland domain (new Fig. S1 in the MS);
- a map showing the GrIS last glacial maximum (LGM) extent simulated by each experiment compared to available LGM reconstructions inferred from proxy records (Evans et al., 2009; Arndt et al., 2015, 2017; Winkelmann et al., 2010) and one modelling study (Lecavalier et al., 2014) (new Fig. 7 in the MS);
- a figure showing the GrIS sea-level contribution for the last 60 ka (new Fig. S4 in the MS).

Regarding the ability in reproducing the NEGIS stream we do not think this is a relevant point for our study, since here we focus on its margin fluctuation, not much on the evolution of the stream itself. Still, this is an interesting detail to show and discuss. To do so, we included:

- a figure showing the simulated and observed PD ice velocities zoomed on the NEGIS sector (new Fig. S2);
- the spatial distribution of the simulated-observed velocities (new Fig. S3, left panel) and a scatterplot between observed and simulated PD velocities (new Fig. S3, left panel) for the NEGIS sector.

Although the magnitude of the ice flow is reasonably well simulated (very fast ice at the margin, that slows down inland), Fig. S2 clearly shows that our model fails in reproducing the detailed present geometry of the ice stream. The ice flows that feed the 79N and ZI are not well reproduced and also the long penetrating tongue of fast ice typical of the NEGIS is simulated as a wider area of fast flow, but not as fast as the observations.

The past evolution of the stream itself simulated by our model is shown in new Fig. 4 (old Fig. 3). However, to our knowledge there are no available proxy data for the past to compare with. New data from ice-core EGRIP project will certainly provide new insights into the paleo evolution of the NEGIS, but the only available information so far is given for the last 400 yr (Vallelonga et al., 2014). Thus, discussing in detail the past evolution of the stream seems to be pointless.

The lack of marine sediment archives in the northeast region hampers the possibility to compare our results with proxy data for the oceanic conditions offshore the NEGIS. This is also the reason why we chose to not include the millennial-scale variability in the oceanic forcing since, when focusing on a restricted area of the ice sheet, very precise information on the oceanic state is required to add a good degree of realism (see a detailed answer to this point below in this document). Estimates of present submarine melting rate at 79N suggest that values of 50 m a$^{-1}$ at the grounding line and 15 m a$^{-1}$, or lower, along the shelf (Wilson et al., 2017). This drop in melt with increasing distance from the grounding line is taken into account by our submarine melt parameterisation. Hopefully new submarine melt estimations in the future will help to better constrain our results.

Nevertheless, we compare our results to other types of reconstruction. Our simulated maximum glacial extent is compared to previous reconstructions based on geological records (Evans et al., 2009; Arndt et al., 2015, 2017; Winkelmann et al., 2010, Funder et al., 2011) showing a fair agreement (new Fig. 7). These and other paleo data investigating the NEGIS margin throughout the last glacial period (LGP) (e.g. Bennike and Weidick (2001), Weidick (1996)) were already considered by Larsen et al. (2018) to construct their grounding-line migration transient profile (Fig 3a of Larsen et al., (2018)). Therefore, by comparing our results to those from Larsen et al. (2018) we are implicitly evaluating our simulated NEGIS front evolution with respect to all these geological records (new Fig. 3, old Fig. 1).

Finally, the transient evolution of the ice flux has been included in the results to help analysing the dynamical effects of the oceanic forcing on the retreat that old Fig. 2 (new Fig. 3 in the MS) may not exhaustively explain. Note that the ice flux has been averaged on the NEGIS sector to smooth the signal. This figure confirms the dynamical reorganization of the NEGIS sector induced by the oceanic forcing.

All the figures cited here have been discussed in the new version of the MS.

Accordingly, we changed the paragraph of Pag. 6 lines 14-19 (old MS) to:

[revised manuscript text omitted]

Another concern is the choice of oceanic forcing applied to the model ice sheet. For simplicity the submarine melt rate is assumed to be spatially uniform around Greenland. Given the lack of data this

can be argued to be a fair assumption. However, the impact of this choice should be documented and discussed in light of existing data from sites along the margins of Greenland. A bigger concern is the inference that past oceanic temperatures below the ice evolve in phase with the atmospheric temperature (eq. 4). Several studies have shown that during glacial periods the subsurface temperatures off Greenland were relatively warm due to the stratification of the water column under an extensive sea ice cover and associated fresh surface layer (see e.g. Alvarez-Solas et al. 2011).

The reviewer is certainly right arguing that subsurface waters of the North Atlantic and Nordic Seas were likely warmer during the stadial phases of the LGP, thus evolving in antiphase with respect to the surface atmospheric temperatures inferred from the ice-core records of Greenland. In another work that investigated millennial-scale variability of the GrIS to abrupt oceanic variations associated with Dansgaard-Oeschger (D-O) cycles (Tabone et al., 2019), we actually forced the ice-sheet model through subsurface temperature anomalies that accounted for this antiphase relation. In that case, the usage of an antiphase signal was highly recommended to correctly interpret the results, since several high-resoluted proxy records from marine sediment cores and modelling studies exist corroborating the hypothesis that subsurface and surface water temperatures were decoupled during D-O events. Thus, that choice was sustained by the short (millennial) timescales involved.

On the contrary, here, we focus on the response of the NEGIS margin to variations in the oceanic temperatures at much longer (orbital) timescales. To make the experiment as simple as possible and within the scope of a sensitivity study, we impose a spatially homogeneous temperature anomaly at the base of the ice shelf that follows climatic variations due to orbital changes. This is based on the assumption that at such long timescales the ocean responds as the atmosphere to changes in insolation, although this is likely a simplification of what probably was the real paleo oceanic circulation around Greenland. Also, due to the lack of proxy data that provide information on oceanic temperature variations that cover large portions of the quaternary (at least for the last 120 kyr), an in-phase ocean-atmosphere relation at long timescales cannot be ruled out (this assumption is actually corroborated by a proxy record for the temperatures of the Arctic Ocean waters at intermediate-depth during the last 50 kyr, as discussed further in this document).

A similar approach has already been successfully used in a previous work (Tabone et al., 2018) that investigated the past role of the ocean on the GrIS evolution during the last two glacial cycles. Of course, a more realistic approach would probably consider submarine melt rates that are depth-dependent (possibly following the currently simulated depth of the grounding line) and that spatially vary across the Greenland coasts, but these are improvements that go beyond our simple sensitivity study. However, these features will be taken into account in a future work for a comprehensive investigation of the problem.

SPECIFIC COMMENTS:

Line 7, page 1: LGP is not a common acronym. Better to spell out last glacial period and if necessary use common acronyms such as the LGM to specify a specific period within the glacial period where appropriate.

Changed accordingly.

Line 14, page 1: NG - a more common acronym for the 79N glacier in the literature is 79N.

Changed accordingly.

Line 16, page 1: it is stated that 79N is more stable than ZI due to its bed configuration - please elaborate on this.

That sentence has been changed to: *"Since 79N is retreating over an upward-sloping bed (Mouginot et al., 2015), it may be less prone than ZI to an unstable retreat. This has been recently examined through an ice-flow model pointing out that its floating tongue has to lose several tens of km of ice before the glacier becomes unstable (Rathmann et al., 2017). A larger stability of 79N has been*

*recently tested under various future warming scenarios by another modelling study (Choi et al., 2017), suggesting that it may be related to the presence of pinning points (such as ice rises) near the calving front. Despite so, the 79N ice shelf is losing mass since 2001 (Mayer et al., 2018)."*

Line 5, page 2: the slow retreat of 79N suggested by Choi et al. is described as conservative. Why? Please elaborate.

This sentence has been changed to : *"A recent study investigating the response of 79N and ZI to oceanic forcing with the aim of constraining their future stability suggests a further slow retreat of 79N and a complete loss of the ZI ice tongue due to increasing melt rates in the next decades (Choi et al., 2017). However, new evidence of retreat of both glaciers beyond their PD margins during the Holocene suggests that the resistance of 79N to increasing basal and frontal melt modelled by Choi et al., 2017 could be too conservative (Larsen et al., 2018)."*

Line 11, page 2: the ice is thought to have retreated 20-40km being its PD position during MIS3. How is this now? Please elaborate and include an assessment of the uncertainties.

This sentence has been changed to: *"The paleo records emerging from this study, combined with a collection of geological data assembled in the last 20 years (Weidick et al., 1996; Bennike and Weidick, 2001; Evans et al, 2009; Winkelmann et al., 2010; Arndt et al., 2015, 2017), suggest that the ice margin considerably fluctuated in magnitude throughout this period. Around 41-26 ka BP during Marine Isotope Stage 3 (MIS-3, c. 60-25 ka) the NEGIS front was ca. 20-40 km farther inland than today, then advanced by more than 250 km toward the shelf break at the Last Glacial Maximum (LGM) and retreated again during the last deglaciation, at ca. 70 km behind its present-day position, where stopped most of the mid-and late Holocene (7.8-1.2 ka BP)."*

Also, added in the Results: *"This reconstruction is a result of averaging the evolution of three NEGIS outlet glaciers fronts (79N, ZI and SG) inferred from the various geological records with respect to their position at 2014 (Howat et al., 2014). Although it is a valuable tool providing a rough idea of the margin fluctuation during the last 45 ka, caution should be taken before performing one-to-one comparisons with model data. Specifically, while the strong retreat during the Holocene is documented for all those glaciers, records showing their margin position during Marine Isotope Stage 3 (MIS-3, ca. 60-25 ka BP) are available only for ZI and SG, which were behind their present location by ca. 20 and 40 km, respectively. However, since they all shared the same behavior during the Holocene, it is likely that 79N front was as inland as the others during MIS-3 (Larsen et al., 2018). "*

Line 24, page 2: resolve "?"

We do not see any "?" at Pag. 2 Line 24 in the published version of the MS under discussion.

Line 8, page 3: the climate forcing is composed of 3 different ice core reconstructions (Vinther, NGRIP, and NEEM). Substantiate why this is done, instead of using only one ice core record such as NEEM.

The choice of a composite climatic index is based on the usage of valuable reconstructions specific for certain intervals of time to provide a final accurate temperature signal valid for the whole domain. Also, temperature reconstructions from NEEM ice core have been found to underestimate the temperature anomaly during some D-O events with respect to those inferred at the dome due to spatial temperature gradients across the GrIS (Guillevic et al., 2013). Although we focus on the NEGIS sector in the analysis, our ice-sheet model is applied to the whole GrIS. Thus, using temperature reconstructions derived from the NEEM ice core only would probably not be the best strategy.

The bulk of the data comes from the well-known work from Kindler et al. (2014) which reconstructed temperatures in Greenland between 120 and 10 kyr from high-resolution $\delta15N$ records from the NGRIP ice core. $\delta15N$ is sensitive to processes of the firn layer and is considered as a good alternative to d18O for inferring surface temperatures. This time series is the longest high-resolution surface temperature reconstruction derived from the NGRIP core so far; calibrated using both proxy

and diffusion models data, it contains all the D-O events of the last glacial, thus it is also appropriate for investigations at millennial timescales. Then, this reconstruction has been expanded to reach the PD and the LIG through other available records. Vinther et al., 2009 provides a temperature reconstruction for the Holocene based on ice cores from Greenland and Canada. This signal accounts for ice-sheet elevation changes occurred during the deglaciation. This is one of the most valuable surface temperature reconstructions available so far for the Holocene (if we do not consider a more recent work from Lecavalier et al., 2017). Finally, temperature reconstructions from 115 to 130 kyr BP are taken from the NEEM ice core (NEEM community members, 2013), since it is the only available ice core as yet that allows to trace Greenland surface temperatures back to the LIG. The final composite serie leads to a temperature reconstruction from which extracting a climatic index to be applied to the whole GrIS.

Line 9, page: why is the variability below orbital removed? What is the purpose of this? What is the model result given the full variability represented by the climate reconstructions? Is there a reason to believe the millennial scale variability should be neglected in forcing the ice sheet?

The millennial-scale variability is here removed to make the message of the paper more direct and straightforward. The experimental design was chosen to be as simple as possible. Focusing on a specific region, such as that of NEGIS, requires a right dose of caution when considering oceanic changes at rapid timescales to represent the behavior of that area from a more realistic point of view. The best way of doing it would be building a signal that comes from proxy records close to the NEGIS area. However, oceanic temperature reconstructions for the northeastern part of Greenland that, moreover, cover a large portion of the LGP do not exist to our knowledge. Considering orbital timescales only in a sensitivity test allows to avoid this issue, leading to a simpler experimental design that directly drives the reader to the message of the work. Nevertheless, records from the Arctic Ocean and Fram Strait suggest that the oceanic conditions considerably varied during the last 50 kyrs. However, the long-scale variation of these temperature records seems to fairly agree with the submarine melting evolution adopted in our work.

To justify this, we added this paragraph in the Discussion: "*Paleoceanographic records inferred from marine sediments in the Arctic Sea and Fram Strait that provide information on the oceanic state during MIS-3 at high temporal resolution are scarce. However, they all suggest rapid temperature fluctuations as a result of large changes in water masses at different depths. Warmer SST may last for 3-4 kyr before cooling (Muller et al., 2014). Generally, strong variations in the oceanic conditions are found between glacial-interglacial, but also between larger stadial-interstadial transitions (Poirer et al., 2012). A sediment record between the Nordic seas and the Arctic Ocean suggest that high SST and low intermediate water temperatures are typical of interstadials, while the opposite is found during stadials due to intrusion of warmer Atlantic subsurface water (Rasmussen et al., 2014). This strong oceanic temperature variability during the last 50 kyr is also documented by another record based on a stack of sediment records of the Arctic Ocean and the Fram Strait, suggesting the occurrence of several peaks of warmings during MIS-3 reaching temperatures 1-3 °C higher than those recorded for the Holocene (Cronin et al., 2012). However, a qualitative analysis of this temperature record at long (orbital) timescales indicate that its evolution agrees well with that of the melting rate signal used in this work: high melting during MIS-3, prolonged cooling during the LGM and high melting again during the Holocene. Thus, even though we remove some degree of realism by not considering the millennial-scale variability in the ocean, our experimental design could fairly represent the evolution of northern Greenland oceanic conditions at long timescales.*"

Line 11, page 3 and eq. 1: PD is referred to as interglacial. Please be more precise on definition of PD: interglacial, Holocene or present day?

PD refers to present day (sensu lato, that is, preindustrial). To make it clearer, this sentence has been changed to: "$T_{LGM, atm} - T_{PD, atm}$ *is the glacial minus present-day (meaning preindustrial) atmospheric temperature anomaly simulated by the climate model of intermediate complexity CLIMBER-3α.* "

Line 13, page 3: why use CLIMBER-3a and not PMIP for the LGM - interglacial climate? What is the impact of the choice of model?

We did not perform simulations with different climatologies since this is beyond the scope of this work. However, we agree with the reviewer that outputs from more recent, comprehensive, and higher resolution climate models might be more appropriate. This will be surely done in our future work. Nevertheless, CLIMBER-3α has been already used in several modelling studies investigating the paleo evolution of past ice sheets proving reliable results (e.g. Alvarez et al. (2011, 2013), Banderas et al. (2012, 2018), Blasco et al. (2019), Tabone et al. (2018, 2019)). Also, since snapshots from CLIMBER-3α are used for defining the transient paleo climatology for the sole atmosphere and since this work focuses on the sensitivity of the NEGIS margin to the ocean, we expect that the choice of the climatology has only second-order effects on the results.

Line 15, eq. 2, page 3: What is the rationale behind choosing the same approach for calculating precipitation as for temperature? Is this appropriate? What is the impact of this choice, please document. Note that P_LGM and P_PD are not described. How are these calculated?

$P_{LGM, ann}$ and $P_{PD, ann}$ are the annual precipitations provided as outputs from simulations performed with the CLIMBER-3α model under weak-AMOC and present-AMOC conditions, respectively (Montoya and Levermann, 2008), as $T_{LGM, atm}$ and $T_{PD, atm}$. Defining the paleo precipitations through an anomaly method with respect to the present, as done with the temperatures, is an approach followed by many models to represent transient past precipitations (e.g., Marshall et al. (2000, 2002), Marshall and Koutnik (2006), Charbit et al. (2002, 2007), Zweck and Huybrechts (2005), Philippon et al. (2006), Colleoni et al. (2014), Banderas et al. (2018)). Note that, here, the ratio of the precipitation anomalies originally comes from the assumption that precipitation depends exponentially on temperature, thus a difference in temperature would result in a ratio in precipitation. Also, it allows to avoid the creation of negative values in the precipitation field, which would not be physical. This approach is especially useful when the ice-sheet model is not coupled to a climate model. Despite its crudity, it allows to better represent past precipitations than directly using paleo records coming from ice cores, since model snapshots describe the spatial precipitation pattern away from the core locations with more accuracy. This method can be then improved for example by incrementing the number of snapshots between the LGM and the PD (Charbit et al., 2002), or correcting the fields with an amplification factor (Banderas et al., 2018). However, for the purpose of investigating the ice-sheet evolution due to oceanic forcing at long timescales the approach used in this work should be sufficiently accurate.

To clarify this point, the sentence at Pag. 3 Line 16 has been changed to: *"The precipitation field is obtained following a similar approach based on the ratio of LGM and present-day precipitations, scaled by α(t), as:*

$$P_{ann}(t) = P_{clim, ann} * (α(t) + (1 - α(t)) P_{LGM, ann}/P_{PD, ann})$$

*where $P_{LGM, ann}$ and $P_{PD, ann}$ are the LGM and PD annual precipitations provided by the same climate simulations of $T_{LGM, atm}$ and $T_{PD, atm}$. This approach has been adopted by many ice-sheet models to represent transient past precipitations when they are not coupled to a climate model (e.g., Marshall et al. (2000, 2002), Marshall and Koutnik (2006), Charbit et al. (2002, 2007), Zweck and Huybrechts (2005), Philippon et al. (2006), Colleoni et al. (2014), Banderas et al. (2018))."*

Line 16, page 3: why use PDD and not scale SMB from MAR which is used for the temperature?

Despite the known shortcomings of the PDD parameterisation, such as its tendency to underestimate the surface melt, especially in warm climates since it does not take into account insolation changes (e.g. Robinson and Goelzer, 2014; Van den Berg et al., 2011), it is still one of the most used approaches in the literature for representing the ablation in paleo modelling (e.g. Peyaud et al., 2007, Born and Nisancioglu 2012, Stone et al., 2013, Quiquet et al., 2013, Banderas et al., 2018). This is because it is a simple method compared to others, such as ITMs (Insolation-temperature melt methods, Robinson et al., 2010) or EBSMs (energy balance and snowpack models, Bougamont et al., 2007), and its results are still rather satisfactory when compared to regional climate models (e.g. Peano et al. (2017), Vernon et al. (2013)). Scaling the SMB directly from MAR would overlook important processes that our ice-sheet model accounts for, such as refreezing of surface waters or the elevation-melt feedback, that can be considered by explicitly calculating the ablation and have

important consequences on the evolution of the ice sheet. The suggested method would probably oversimplify the experiment for our purposes.

Line 17, page 3: it is claimed that using PDD does not "jeopardise" results as focus is no oceanic forcing. Note that this invalidates any comparison of the relative importance of atmospheric and oceanic forcing. Please elaborate on this point + check manuscript for consistency with the discussion of the importance of the oceanic forcing given that its relative role cannot be assessed.

The reviewer is right arguing that the relative role between atmospheric and oceanic forcings in the NEGIS retreat during MIS-3 cannot be determined from our results due to the lack of a sensitivity test on the atmosphere. However, assessing this is beyond the scope of our work, which simply aims to show the potential of the oceanic forcing in driving the margin of the NEGIS throughout the last glacial period.

To make it clear, this paragraph has been changed to: *"Surface ablation is calculated by the simple positive degree (PDD) scheme (Reeh, 1989). Although this method does not account for past insolation changes, and since here we primarily investigate the sensitivity of the NEGIS to the oceanic forcing during glacial times, the choice of this melt scheme instead of another should bring only second-order effects to the overall results of this work."*

Also this paragraph has been added in the Discussion: *"Further experiments accounting for changes in the atmospheric temperatures and precipitation or variations in the external forcing (i.e. insolation) should be carried out for a full understanding of the mechanisms involved in this retreat, here explained by considering the sole impact of the ocean. Particularly, a sensitivity study on climatic variations performed with a prescribed ocean could help constraining the effect of the atmosphere in this phenomenon to eventually evaluate the relative role between the forcings in driving the NEGIS margin."*

Line 8, page 4: resolve "?"

We do not see any "?" at Pag. 4 Line 8 in the published version of the MS under discussion.

Figure 3: what is shown here. Please specify time periods for each subplot.

New Fig. 4 (old Fig. 3) now shows the time period for each subplot.

Figure 4: Show A and B in relation to ice margin (e.g. in figure similar to 3). Specify where smb and Bmelt are taken from.

Now the inset map of Fig. 6 (old Fig. 4) shows the maximum glacial extent for the experiment κ=8 m a$^{-1}$ K$^{-1}$. Surface mass balance and basal melt are simulated by the model and averaged over the regions A and B.